# Multicomponent dynamics in amorphous ice studied using X-ray photon correlation spectroscopy at elevated pressure and cryogenic temperatures
Aigerim Karina [1,13], Hailong Li [2,3,13], Tobias Eklund [2,4,5], Marjorie Ladd-Parada [6], Bernhard Massani [7], Mariia Filianina [1], Neha Kondedan [1], Andreas Rydh [1], Klara Holl [2,4], Ryan Trevorah [8], Simo Huotari [8], Robert P. C. Bauer [9,10], Claudia Goy [9], Nele N. Striker [9], Francesco Dallari [9,12], Fabian Westermeier [9], Michael Sprung [9], Felix Lehmkühler [9,11] & Katrin Amann-Winkel [1,2,4] ✉

Knowing the pressure dependence of glass forming liquids is important in various contexts. Here, we study the case of supercooled water, which has at least two different amorphous states with different densities. The pressure dependencies of the two glass transitions are predicted to show opposite behaviour, crossing in the P-T plane at elevated pressure. The experimental identification of the glass transition at elevated pressure and cryo-conditions is technically difficult. Moreover, in the case of amorphous ices, the glass transition is interrupted by crystallization, which makes it even more challenging. We show the feasibility of performing X-ray photon correlation spectroscopy experiments at elevated pressure using a diamond anvil cell at cryogenic temperatures. We observe two dynamic components when approaching the glass transition temperature. For high-density amorphous ice at a pressure of around $(0.08 \pm 0.02)$ GPa we determine the glass transition to be at higher temperatures compared to ambient conditions.

Amorphous ices are solid states of water which lack long-range order. At least two distinct types of amorphous ices are known: low- and high-density amorphous ices (LDA, HDA)[1]. Such polyamorphism of water could potentially explain water's anomalous properties even at temperatures above 0 °C, like the density maximum at 4 °C, based on the two-states model of water proposed in the early 1990s[2]. The model suggests the existence of two liquid states of water, low- and high-density liquid (LDL, HDL), at deep supercooled temperatures and slightly elevated pressure. These liquids are connected through a glass-liquid transition to the amorphous ices LDA and HDA. The anomalous behavior of water is related to a critical point at temperatures around 230 K and pressure over 1 bar, and fluctuations in the region above this critical point[3].

Furthermore, experiments and theory both have found that water behaves differently as simple liquids, regarding its compressibility, heat capacity, glass transition, viscosity, etc. For most liquids, the glass transition temperature ($T_g$) and viscosity ($\eta$) tend to increase with higher pressure values. Not only does water have two glass transitions (with respective $T_g$)[4], it is also remarkable that the respective transition temperatures show an almost opposite pressure dependence leading to a crossing at elevated pressures, somewhere below 0.3 GPa[5]. The pressure dependence of water's $T_g$ is derived from very few experimental data points[4,6,7], combined with computational studies[8]. The lack of experimental data is due to the difficulty of using existing experimental techniques at elevated pressures. The crossover was found by recovering the amorphous states at ambient pressure and liquid nitrogen temperatures. Consequently, HDA can be studied even in

[1]Department of Physics, Stockholm University, Stockholm, Sweden. [2]Max-Planck-Institute for Polymer Research, Mainz, Germany. [3]State Key Laboratory of Fine Chemicals, School of Chemical Engineering, Dalian University of Technology, Dalian, China. [4]Institute of Physics, Johannes Gutenberg University Mainz, Mainz, Germany. [5]European X-ray Free-Electron Laser, Schenefeld, Germany. [6]Department of Chemistry, Glycoscience Division, Stockholm, Sweden. [7]The University of Edinburgh, School of Physics and Astronomy (SoPA), Centre for Science at Extreme Conditions (CSEC), Edinburgh, UK. [8]Department of Physics, University of Helsinki, Helsinki, Finland. [9]Deutsches Elektronen-Synchrotron DESY, Hamburg, Germany. [10]Freiberg Center for Water Research, Technische Universität Bergakademie Freiberg, Freiberg, Germany. [11]Hamburg Centre for Ultrafast Imaging, Hamburg, Germany. [12]Present address: Department of Physics and Astronomy, University of Padova, Padova, Italy. [13]These authors contributed equally: Aigerim Karina, Hailong Li. ✉e-mail: amannk@mpip-mainz.mpg.de

the stability region of LDA. While HDA's glass transition temperature shows a positive slope with pressure, LDA's $T_g$ decreases with increasing pressure[5]. In the same pressure range the melting curve of water[9] shows a negative slope, with a minimum in the $I_h$-III-liquid triple point at ~0.2 GPa; whether or not these phenomena are connected remains unclear. Liquid water's viscosity as a function of pressure[10,11] has a negative slope up to around 0.2 GPa[11] and a subsequent positive slope at higher pressures, while this minimum in the pressure dependence is most pronounced at supercooled temperatures.

The nature of the glass transition in supercooled liquids, not only water, is highly debated and different mechanisms are discussed in literature[12,13]. A combination of different tools and approaches is needed to shed light on this debate. Most experimental methods used to investigate the glass transition measureing changes in heat capacity (calorimetry), relaxation and rotation (dielectric spectroscopy, NMR, light scattering), or volumetric changes (dilatometry). X-ray photon correlation spectroscopy (XPCS) is a powerful tool to investigate collective dynamics in glass-forming systems on a timescale of microseconds to minutes (or hours)[14]. XPCS makes use of coherent X-ray radiation—available only at modern synchrotron radiation and free-electron laser (FEL) facilities—and studies the time evolution of the speckle pattern originating from interferences of the coherent X-ray beam scattered by the disordered molecules in the sample. XPCS is the X-ray analogue to dynamic light scattering (DLS), suitable for measuring diffusion processes in transparent samples down to microsecond timescales. We have previously successfully applied XPCS to measure the dynamics of amorphous ice at ambient pressure conditions, demonstrating a crossover in the dynamics at the same temperature where calorimetry observes an increase in heat capacity[15]. We relate these dynamics to mainly translational motion of water molecules, as the photons couple to the electron cloud and are not sensitive to rotational motion. In addition, we recently investigated a free-standing ice layer in vacuum, where a heterodyne signal was detected in XPCS measurements, which is presumably related to LDA regions "floating" inside a HDL matrix[16,17]. This would only be possible due to different glass transition temperatures at ambient pressure. Referring to structural data, in recent

years new in situ studies using X-ray scattering during compression and decompression of amorphous ice inside a diamond anvil cell (DAC) discussed their relation to the corresponding liquid states[18,19]. Experimental evidence of low-density liquid water was found upon rapid decompression[20]. In addition, relaxation dynamics at high pressures have been measured via XPCS only recently for metallic glasses[21,22] also using a DAC, as well as gelation of a protein solution using a hydrostatic pressure cell[23], both at room temperatures or above.

In this work, we investigate the dynamics of equilibrated high-density amorphous ice (eHDA) at elevated pressure using XPCS. We use ex-situ prepared samples re-pressurized in a DAC. Setup and detector arrangement are shown in Fig. 1. We demonstrate that it is possible to combine XPCS with a DAC inside a cryostat, and follow the dynamics in eHDA during heating at an elevated pressure of around 0.08 GPa. The error in pressure determination is at least ± 0.02 GPa, due to limitations of the used method (see SI for details). We further discuss the experimental challenges related to this approach.

## Results and discussion

We study the transition in amorphous ice by increasing increments of temperature from high-density amorphous ice (HDA) at elevated pressures. Collective dynamics of the water molecules are investigated by following the intensity fluctuations of the coherent diffraction signal. In total, we measured three samples and determined the transition temperature at different pressures. The XPCS analysis focuses on samples A and B, because Pseudo-Kossel lines (streaks in small-angle X-ray scattering [SAXS] image in Fig. 1A) become stronger at higher pressures and overlap with the speckles on the detector (see also SI, Table S1 and Fig. S3). Here, we compare the two samples measured at a pressure of $(0.08 \pm 0.02)$ GPa at different flux densities. In Fig. 2A the wide-angle X-ray scattering (WAXS) signal of sample A is shown during heating. At the lowest temperature (93 K) it shows the characteristic halo peak of HDA centred around 2.1 Å$^{-1}$. At 114 K the sample transforms to LDA resulting in a shift of the diffraction signal to lower Q. While the diffraction pattern of the first measurement at 114 K

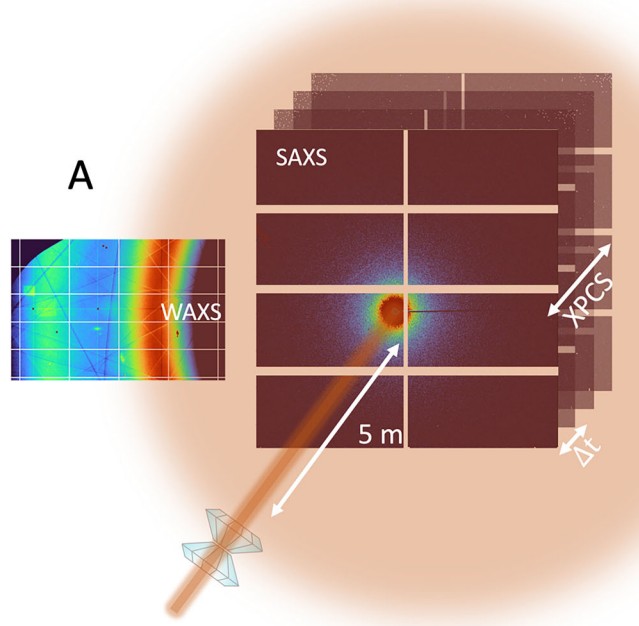

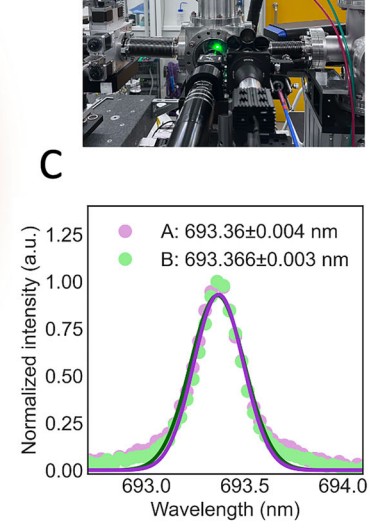

**Fig. 1 | Details of the high-pressure XPCS experiment under cryogenic conditions. A** Experimental setup for XPCS in SAXS (small-angle X-ray scattering) geometry with simultaneous WAXS (wide-angle X-ray scattering) measurements. In both detector images Pseudo-Kossel-lines are visible, which are caused by the

diamonds[32]. **B** Customized DAC-cryostat setup at beamline P10 (PETRA III). **C** Ruby fluorescence spectra of samples (**A**, **B**) at 93 K, Gaussian fits to data points are drawn as solid lines.

**Fig. 2 | Results from sample A (eHDA) at ~0.08 GPa at different temperatures.** The given temperatures reflect the sample temperature (see SI). **A** WAXS I(Q), **B** SAXS I(Q), both averaged over the first 300 s of a measurement series. In (**C**) the intermediate scattering functions calculated at Q = 0.0033 Å⁻¹ are shown, data are vertically offset for clarity. **D** Relaxation time of the exponential decay as a function of temperature. Data was taken at a photon flux density of $1.3 \times 10^7$ photons/(s μm²).

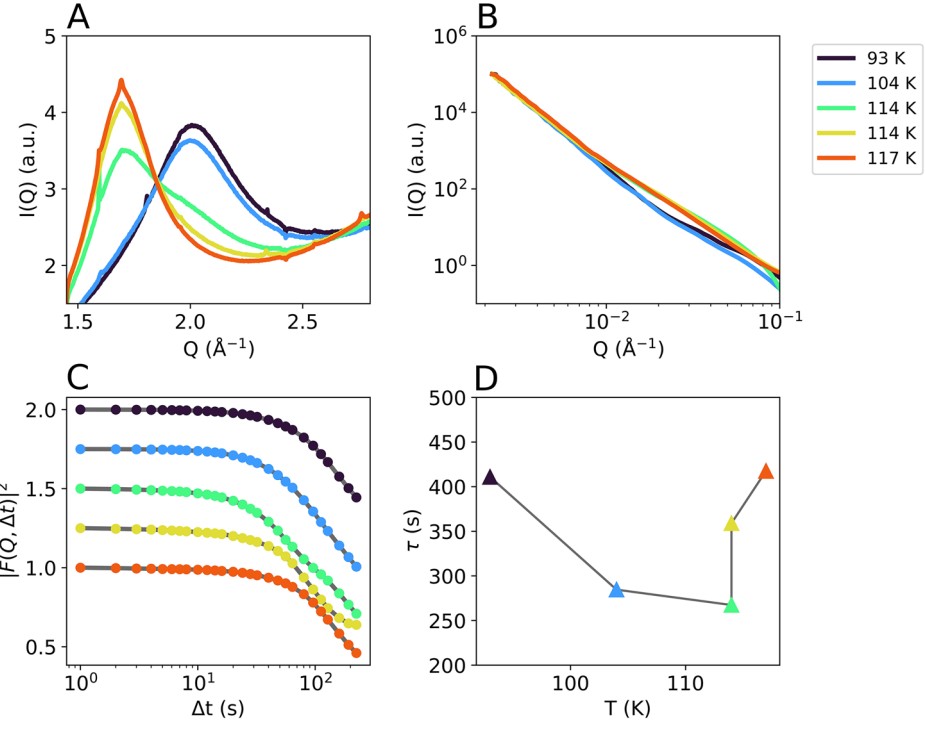

shows a coexistence of eHDA and LDA, the transformation is complete at the second dataset (also 114 K). The WAXS (Fig. 2A) and SAXS (Fig. 2B) I(Q) data are averaged over the first 300 s of a series, since temporal heterogeneities have been observed at longer timescales (see SI Fig. S9). The curve shape of the SAXS signal changes with the appearance of LDA at 114 K. We further calculate the temporal intensity autocorrelation function $g_2$ as previously described[24,25]. The software is based on a multi-tau algorithm thatnaturally results in an almost logarithmic binning of the $\Delta t$ values.

$$g_2(Q, \Delta t) = \frac{\langle I(Q, t)I(Q, t + \Delta t)\rangle}{\langle (I(Q, t))\rangle^2} \quad (1)$$

Here, I(*Q, t*) is the scattered intensity at time *t* and I(*Q, t* + *Δt)* is the scattered intensity after a lag time $\Delta t$. The averaging is done over all times *t* and over all pixels of the detector within a certain *Q*-range. ($g_2$ - 1) is proportional to the square of the intermediate scattering function $|F(Q, \Delta t)|^2$, with $\beta$ being the speckle contrast.

$$g_2(Q, \Delta t) = 1 + \beta |F(Q, \Delta t)|^2 \quad (2)$$

The computed $|F(Q, \Delta t)|^2$ functions are analysed by fitting a distribution of exponential functions with a Kohlrausch–Williams–Watts (KWW) exponent $\gamma$ to the data.

$$|F(Q, \Delta t)|^2 = e^{-2(\frac{\Delta t}{\tau})^\gamma} \quad (3)$$

where $\tau$ is the relaxation time. Stretched correlation functions ($\gamma$<1) are typically found in supercooled liquids, while Brownian diffusive processes are characterized by $\gamma$ = 1. Glasses usually show compressed exponentials ($\gamma$>1), usually assigned to stress relaxation phenomena[14,24]. Please note, that the here derived relaxation time is measured in SAXS geometry at a momentum transfer range of 0.002 Å⁻¹ < Q < 0.02 Å⁻¹ and hence probes a collective motion at length scales from 10 nm to several 100 nm rather than the molecular self-diffusion. A relaxation time of 100 s therefore does not automatically relate to the glass transition, as known e.g., from dielectric

measurements. In our analysis we relate the glass transition temperature to the above described KWW exponent $\gamma$ as well as to the Stokes-Einstein diffusion coefficient[14,25] D, following our previous approach at ambient pressure[15–17].

In contrast to the temporally averaged g2 function, the two-time intensity correlation (TTC) function can resolve temporally heterogeneous dynamics by providing the temporal evolution of the intensity correlation[24,26].

$$C(Q, t_1, t_2) = \frac{\langle I(Q, t_1)I(Q, t_2)\rangle_{pix}}{\langle I(Q, t_1)\rangle_{pix}\langle I(Q, t_2)\rangle_{pix}} \quad (4)$$

Within the TTC function $C(Q, t_1, t_2)$ the $\langle \ldots \rangle_{pix}$ denotes averaging over pixels within a certain Q-range.

The intermediate scattering functions $|F(Q, \Delta t)|^2$, derived from g2 (Eq. 2) of the corresponding runs are shown in Fig. 2C for a momentum transfer $Q = 0.0033$ Å⁻¹. Correlation functions were calculated as well for the first 300 patterns of a series. Data are normalized to the speckle contrast value measured from a static aerogel sample prior to the experiment. We observe the decay of $|F(Q, \Delta t)|^2$ getting faster while heating eHDA until its transformation and a subsequent slowing down with the formation of LDA. The Q-dependent relaxation rate $1/\tau$ was obtained from fitting a KWW function (Eq. 3) to the data, plotted for different temperatures at $Q = 0.0033$ Å⁻¹ in Fig. 2D. The correlation functions and related fits are shown in the SI (Fig. S8). The temperature dependence of the relaxation time demonstrates an initial acceleration and subsequent slowing down of the motion when the sample has transformed to LDA.

To understand the time-dependence of the dynamics in sample A, we looked at two-time correlation (TTC) maps, calculated according to Eq. 4 for the first 300 s of a series (Fig. 3). TTCs for 1000 s are shown in the SI (Fig. S9). While the sample shows mostly stable dynamics at 93 K (Fig. 3A) throughout the 300 s, at 104 K (Fig. 3B) an acceleration of the dynamics is visible over time, as indicated by a reduced width of the diagonal line[24,26,27]. This acceleration is not related to major structural changes, as the WAXS signal is constant during this time interval of 300 s (Figs. 2A and S7). At 114 K (Fig. 3C), with the partial appearance

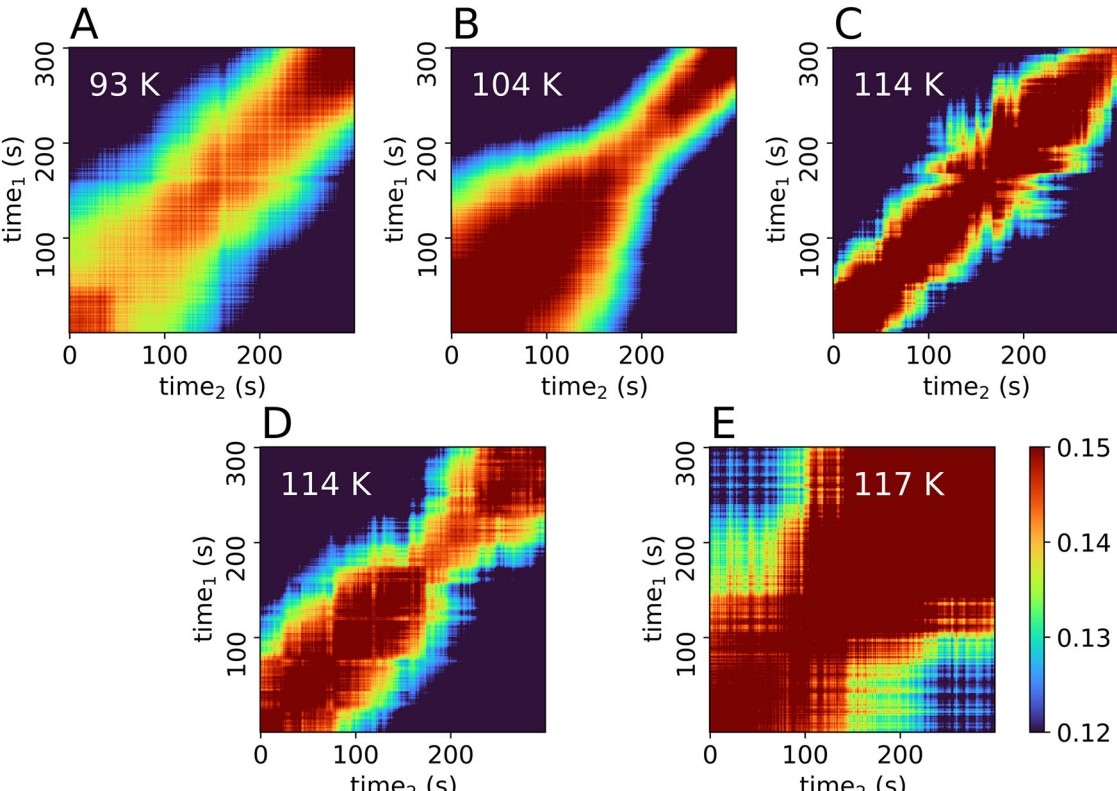

**Fig. 3 | Two-time correlation (TTC) maps of sample A at four different temperatures, calculated for 300 s at $Q = 0.0033$ Å$^{-1}$, at a pressure of ~0.08 GPa.** TTCs for 1000 s are shown in the SI. The given temperatures reflect the sample temperature (see SI), with stable dynamics at 93 K (**A**) and an acceleration at 104 K (**B**). Two consecutive measurements were taken at 114 K (**C, D**). At 117 K the sample fully transformed to LDA (**E**), dynamics are slower.

of LDA (green line in Fig. 2), the dynamics of the sample are overall faster and display only minor heterogeneities over time. For the second run at 114 K (Fig. 3D) the TTC is slightly slower but similarly heterogeneous, while Fig. 2A shows a clear LDA signal (yellow line) at this temperature. Lastly, at 117 K (Fig. 3E) the transformation to LDA seems to be almost finished and thus the dynamics become much slower, especially after the first 100 s.

A second sample spot within sample A at 104 K (T$_{sample}$) was measured (Fig. 4), where the sample transforms from HDA to LDA during the 1000 s duration of the measurement (Fig. 4A). The transition is clearly visible by the shift of the first diffraction maximum from 2.1 Å$^{-1}$ to 1.7 Å$^{-1}$ in the WAXS I(Q). At this low temperature, eHDA is usually expected to not transform to LDA within such a short timescale[28]. The evolution of the diffracted intensity over time is displayed in a 2D map in Fig. 4B. The HDA to LDA transition is reflected in a dynamical crossover throughout the 1000 s and can be derived from the TTC (Fig. 4C). The TTC is derived for a $Q$ of 0.0035 Å$^{-1}$. Since the structure changes over time, we calculated the intermediate scattering function (Fig. 4D, F) for the first and last 200 s, respectively. During this time interval the WAXS intensity was constant, hence no major structural changes are observed, which allowed us to determine the relaxation dynamics of HDA at a temperature of 104 K during the first 200 s. The derived relaxation time as a function of momentum transfer follows a linear Q-dependence, characteristic for ballistic (non-Brownian) motion (Fig. 4E). However, the linear behaviour exhibits an offset, which implies that there might be another process involved and the data might not be fully explained by a single exponential decay. After the transition to LDA the dynamics clearly slowed down (Fig. 4F). This effect was already observed for such samples measured at ambient pressure[15]. However, the transition temperature itself is unusually low, compared to our previous measurements at ambient pressure. We therefore measured a second sample (sample B) using a lower X-ray flux density.

For sample B (Fig. 5), we used a photon flux density of $6.5 \times 10^6$ photons per s μm$^2$ i.e., half of the flux density used for sample A ($1.3 \times 10^7$ photons per s μm$^2$). As previously stated, the pressure for both samples was almost identical (~ 0.08 GPa). The diffraction maximum of the WAXS signal shows a shift from HDA to LDA(L), followed by crystallization (Fig. 5A). The higher thermal stability of sample B compared to sample A is obvious, as the transition to the low-density state now takes place at temperatures above T$_{sample}$ = 123 K, consistent with literature data[28]. The crossover is also visible in the SAXS I(Q) (Fig. 5B). The corresponding correlation functions over 1000 s are plotted for each temperature in Fig. 5C. As for the previous data of sample A, a faster decay can be observed with increasing temperature, followed by a slow-down when the sample transforms to LDA(L) (orange curve at 133 K). The red curve at 143 K shows a multi-step decay, which might be caused by the onset of crystallization, the related TTC shows a deceleration over time (Fig. S12). Figure S12 in addition shows the XPCS analysis after the transition to LDL or LDA and ice I. At 128 K the data let us assume a liquid-like character of the sample (LDL). A separate measurement on pure LDA over the whole temperature range would be necessary to confirm the location of the glass transition in LDA and the here observed liquid-liquid transition free of doubt. A detailed discussion of the LDL state can be found in the SI. We here focus on the analysis of the diffusive process related to the formation of HDL.

Figure 6 shows the relaxation data of sample B, related to the four lowest temperatures where the sample is still eHDA, based on the WAXS intensity (Fig. 5A). We observe an acceleration at times > 500 s (TTCs in Fig. 6), therefore the relaxation rate was calculated only for the first 500 s. Once more, we observe a clear acceleration of the dynamics at higher temperatures. The correlation functions (Fig. 5C) can be described by a KWW function (Fig. S11) similarly to sample A; however, this describes the data with restrictions. At low temperatures a weak Q-dependence is visible, while the Q-dependence at higher temperatures is clearly offset from the origin (Fig. S11).

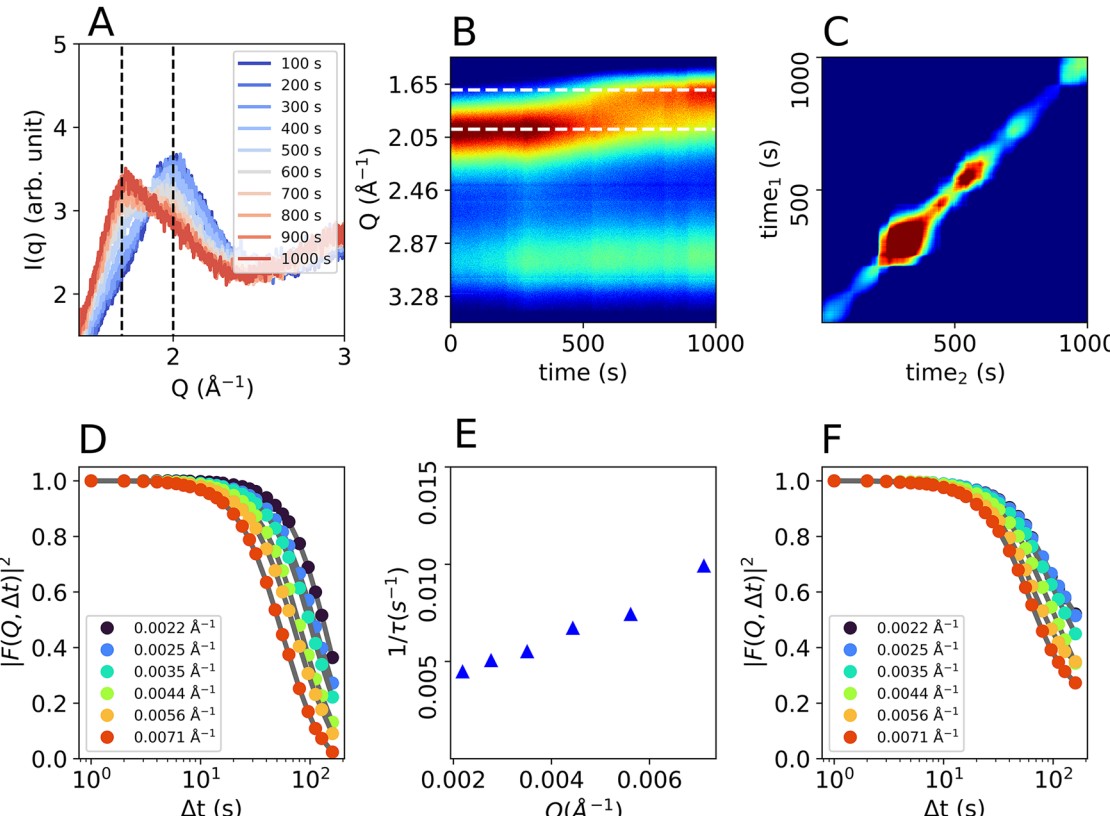

**Fig. 4 | Results from sample A measured at a second spot at 104 K (T_sample) and ~0.08 GPa. A** Integrated WAXS intensity. **B** Intensity map showing the evolution of the diffraction maxima during heating, displaying high intensity with red and low intensity with blue. **C** TTC over 1000 s displaying heterogeneous dynamics during heating. **D** Correlation functions calculated from the first 200 s and (**E**) the corresponding relaxation rates. **F** Correlation functions calculated from the last 200 s.

Some correlation curves cannot be described by a single stretched or compressed exponential function, as given in Eq. 3. There is at least a second dynamical component present, the origin of the different components remains unclear. In order to shed light on the underlying processes, we modelled the intermediate scattering function (see $F(Q, \Delta t)$ in Eq. 2) as a linear combination of a ballistic and a diffusive component:

$$g_2(Q, \Delta t) = b(Q) + \beta(Q) \cdot \left( A e^{-(\nu Q \Delta t)^2} + (1 - A) e^{-(D Q^2 \Delta t)^\gamma} \right)^2. \quad (5)$$

The diffusive term features a KWW-exponent, allowing for a stretched exponential decay ($0 < \gamma < 1$), while the ballistic term is a purely Gaussian decay[29]. The parameters $b$ and $\beta$ are Q-dependent to account for shifted baselines and early decays that are not captured well at our 1 Hz frame rate. As this model postulates a specific Q-dependence, it is fitted to curves for all Q-values simultaneously. This results in a single best fit velocity ($\nu$) and diffusivity ($D$) per experimental run (similar to Provencher's and Štěpánek's MULTIQ approach[30]). Note that we did not—as is commonly done—omit the cross-term that arises when expanding the squared expression. Additional information on this model, the fitting procedure and their parameters is available in SI (Fig. S13). Figure 6 (lower row) shows the related fits for 117 K to 123 K. At the lowest temperature (93 K) this model gives a poor overall fit (not shown). With increasing temperature, we observe that the diffusion coefficient ($D$) rises sharply (i.e., the decay rate increases) between 120 K and 123 K, while the ballistic velocity ($\nu$) remains almost constant (Fig. 6, lower right panel).

## Conclusion

Water´s phase diagram is extraordinarily complex, leading to the ongoing debate of whether or not the amorphous states are connected to their liquid counterparts, and at which temperature and pressures the glass transition

appears. In total, we cryo-loaded three different eHDA samples to a DAC, the results are summarized in Fig. 7. The transition to the low-density state LDA(L) (red-shaded area) at lower pressures and crystallization to ice IX at higher pressure (black triangle) defines the area where eHDA is metastable (blue shaded area in Fig. 7). We see a clear pressure dependence of the crystallization temperature, which is in good agreement with literature[31].

With the goal to determine relaxation dynamics at cryogenic temperatures and elevated pressure, we combined XPCS with a DAC. Of the three samples loaded, we were only able to study the intrinsic dynamics of two samples (A, B) at ~0.08 GPa using XPCS, as at higher pressures we observed strong Pseudo-Kossel lines presumably caused by the used diamonds of type Ia (see also SI)[32]. This evidences that one of the challenges of the method is the choice of diamond type and thickness, to avoid scattering background effects and X-ray absorption. Still, the XPCS method bears a high potential. In this first study, the effect of X-ray induced dynamics is also discussed.

We used different X-ray fluences, and observed at the higher photon flux density only one dynamic component, while at lower photon flux density two dynamic components appear when approaching the glass transition temperature. For sample A, measured with a higher photon flux density, the phase transition from HDA to LDA appears unexpectedly at around T_sample = 114 K (red square), additionally the relaxation rate shows a linear Q-dependence, indicative for a ballistic, non-Brownian motion. The observation can be interpreted as an X-ray induced transition, although other factors, like the thermal history, cannot be fully excluded. We conclude that the phase transition from HDA to LDA has been accelerated by the X-ray flux, as the sample transforms at a temperature more than 10 K lower than sample B. Such acceleration effects, without structural damage, have also been observed in hydrated proteins[33], where normalizing the $g_2$ time axis with respect to the flux density was possible, but cannot be applied here, due to the narrow temperature window. For sample A, once, the

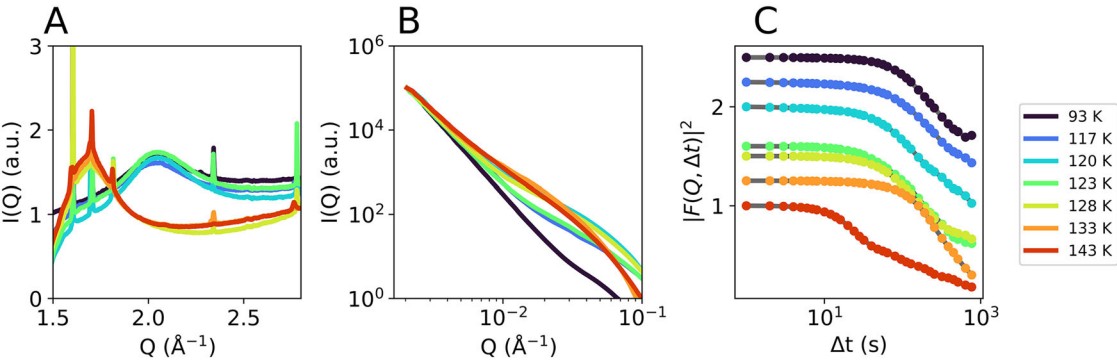

**Fig. 5 | Results from sample B (eHDA) at ~0.08 GPa, using a photon flux density of $6.5 \times 10^6$ photons per (s μm²). A** WAXS and **B** SAXS I(Q), both averaged over 1000 s. **C** Correlation functions at $Q = 0.0035$ Å$^{-1}$.

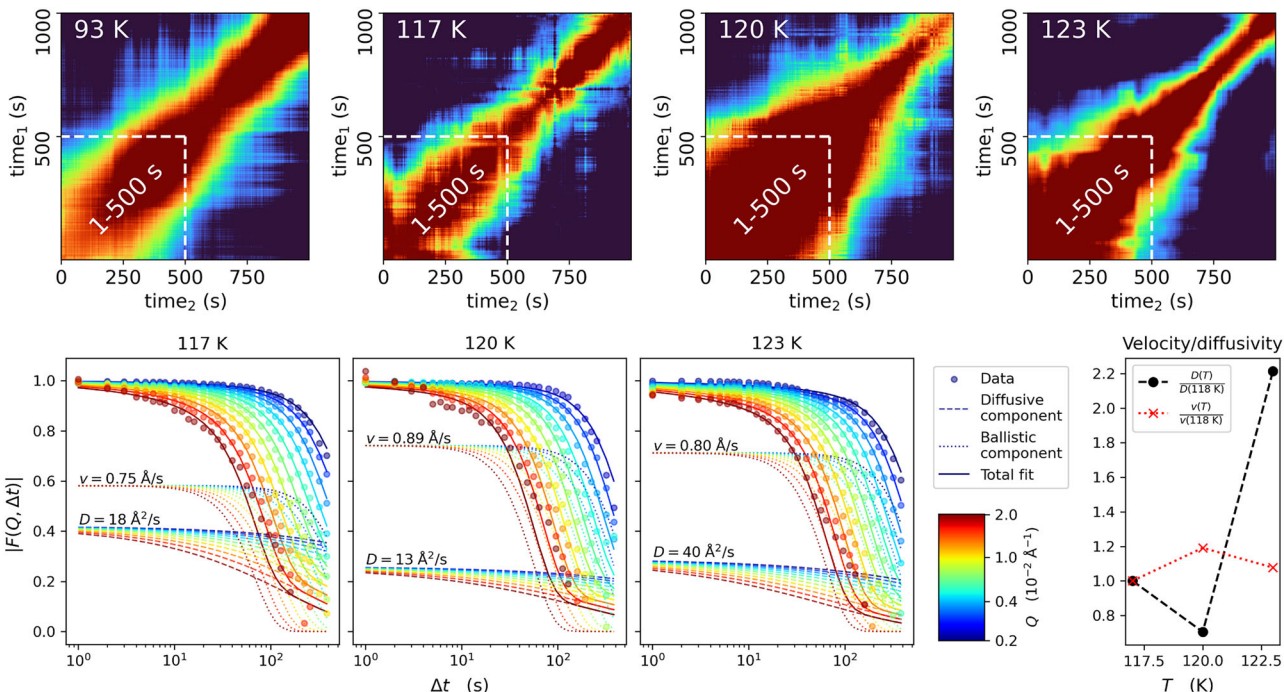

**Fig. 6 | Results for sample B.** Upper row: TTCs for 4 different temperatures (color code as in Fig. 3); Lower row: The corresponding correlation functions calculated for the first 500 s and the relaxation times derived from $F(Q, \Delta t)$ by a multi-fit approach, simultaneously fitting (solid lines) all Q-values in the range 0.002 Å$^{-1}$ < Q < 0.02 Å$^{-1}$ with a linear combination of a ballistic (dotted lines) and a diffusive component (dashed lines). Extracted velocities $v$ and diffusion coefficients $D$ are shown as a function of temperature in the lower-right panel.

complete transformation even occurred within the 1000 s duration of a measurement at a constant temperature of 104 K. In contrast, the sample measured at a lower photon flux density (sample B) showed a higher thermal stability, as the transformation occurred at around T$_{sample}$ = 123 K. Still, we observed in the two-time correlation maps an acceleration of the dynamics at longer timescales. If a diffusive motion is present, $1/\tau(Q)$ would show a linear relationship with $Q^2$ by $1/\tau(Q) = D Q^2$, where $D$ represents the Stokes-Einstein diffusion coefficient[14,25]. By simultaneously fitting the sum of two components to all Q-values of the $g_2$ functions, namely a diffusive and a ballistic component, we observe that during heating of HDA the diffusivity increases. While the velocity of the ballistic component remains almost constant (Fig. 6), the strength of the ballistic component grows. The nature of the ballistic component can be manifold. One possible explanation relates to earlier findings at ambient pressure, were the ballistic component seems to be related to the growth of static LDA domains, as the transition to LDA also involves a volume change of almost 20%[16]. Strain release or growth of crystalline domains might be another origin. While the domains grow in

size, this might explain the increasing strength of the ballistic component with constant velocity. At a sample temperature of 123 K, the WAXS pattern of sample B (Fig. 5A) shows that the sample does not transform to LDA within the investigated 500 s, but crystalline Bragg peaks appear. At this temperature the relaxation dynamics become dominated by Brownian motion, causing the faster decay. The diffusion coefficient $D$ is found to be 40 Å²/s, which must be related to the appearance of an ultraviscous component, HDL. This value of $D$ is similar to our results measured previously at ambient pressure in the vicinity of the glass transition temperature[16]. Based on this finding, T$_g$ at ~0.08 GPa was estimated to be at 123 K (black spot, Fig. 7). To summarize, sample A follows a direct (presumably beam driven) transition HDA → LDA, while sample B undergoes the pathway HDA → HDL → LDL/LDA. Literature as well as the here presented data, let us assume, that sample B proceeds through a liquid-liquid transition (HDL → LDL) at this pressure and temperature range. Our data demonstrate that the glass transition temperature T$_g$ for HDA, as measured by XPCS, has a positive pressure dependence. This is consistent with other findings in

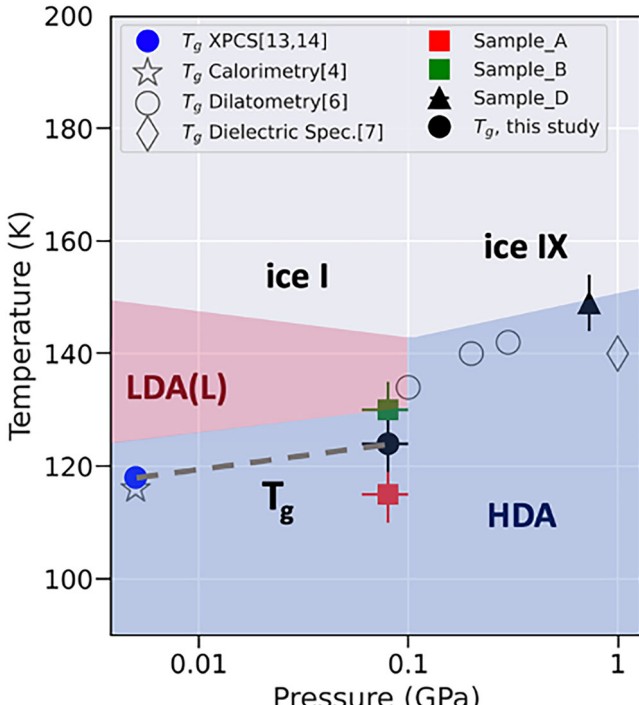

**Fig. 7 | A schematic phase diagram showing the transition- and crystallization temperatures of three eHDA samples measured at different pressures.** Sample A (red) and B (green) transform predominantly to LDL (and ice I). Sample D (black triangle) crystallizes directly to ice IX. From sample B, $T_g$ was determined by XPCS at ~0.08 GPa and 123 K (black spot). Other literature values for the glass transition $T_g$ are shown for comparison, determined by different experimental methods as cited in the references.

literature, which determined the glass transition temperature from volumetric measurements using a piston cylinder setup[6], dielectric spectroscopy[7] and MD simulations[8]. The experimental data are compared directly within Fig. 7 (open circles and open diamond, respectively). The offset between the datasets, e.g., to the volumetric data, is well-founded by the distinct measures, different pathways (isobaric vs. not), heating rate and corresponding crystallization temperatures. Despite the offset, all methods show the same trend, demonstrating that the glass transition temperature increases with pressure.

## Methods

### Sample preparation
Equilibrated high-density amorphous ice samples (eHDA) were prepared on a Zwick Z-100 mechanical press at Stockholm University (for details see refs. 34,35). Five hundred microliter of ultrapure deionized water (Milli-Q water) were loaded into a cylinder-shaped indium container and mounted in a liquid nitrogen (LN₂) precooled piston-cylinder setup. The sample was compressed at 78 K in a LN₂ bath to form unannealed high-density amorphous ice (uHDA)[36], which was then annealed at 160 K under constant pressure of 1.1 GPa to a more densified form of HDA and finally decompressed at 140 K. The eHDA sample was quench-recovered at 0.08 GPa and stored in a LN₂ dewar. Just before the XPCS experiment, a small fraction of the sample was carefully sliced and slightly ground under liquid nitrogen to produce a small piece of ice, and then transferred into the sample cavity of a pre-cooled diamond anvil cell, while keeping the temperature below 100 K, as described below. All samples have been prepared following the same protocol.

### Diamond-anvil cell (DAC) setup
The DAC setup consists of a cryostat, a ruby fluorescence system, and the DAC (Fig. S3). The symmetric DAC contains two opposite seated diamonds

(Boehler-Almax Design, purchased from Almax EasyLab) with a thickness of 1 mm each and a culet diameter of 800 μm. This rather thin and flat diamond design allows for a high X-ray transmission at a pressure range of up to 2–3 GPa. A 250 μm thick steel gasket, pre-indented to around 60 μm, was used to confine samples in a cylindrical chamber with a 500 μm hole.

### Cryo-loading procedure
As a pressure marker, a ruby sphere (BETSA, ~5 μm) was used[37,38] and loaded into the cell at room temperature after which the DAC was cooled to 77 K in LN₂. The eHDA sample was ground in a LN₂ bath. A single small piece of ice was placed inside of the gasket hole onto the lower piston of the DAC. Both parts of the cell and the gasket were pre-cooled in a LN₂ bath prior to mounting. After carefully covering the sample with the upper part of the cell the screws were tightened. The DAC was mounted on a customized copper holder by sliding it into the Cu-block (Fig. S3-B₁) inside a LN₂ bath. This might lead to a positional uncertainty of ±1 mm. The Cu-block was screwed to the cold finger of a cryostat, which was subsequently inserted into the vacuum chamber at the beamline (Fig. 1B).

### Cryostat and sample temperature
We used a JANIS ST-400 cryostat (run with LN₂). Two Si-Diodes are placed at the bottom of the cold finger, one buried inside the cold finger close to the heater element (Diode A), as provided by the company, and a second Diode B mounted on the copper holder (Fig. S3). Both diodes were used to monitor the temperature throughout the experiments. Due to the current cryostat / sample-holder design, a small gap in the order of several 100 micrometers remains between the Cu-sample holder and the DAC (Fig. S3-B₁), which cannot be avoided when cold loading the DAC (Fig. S3-B₁). The temperature of the sample is therefore consistently higher than the reading of the two diodes. We estimate an offset of about 12–15 K between the measured temperature (Diode A) and that of the sample inside the DAC by using an additional thermocouple. Temperatures given throughout the manuscript are offset-corrected. Details to this temperature calibration are described in the SI, see Figs. S4 and S5 and Table S2.

### Pressure determination
We measured ruby fluorescence spectra at 93 K (sample temperature) prior to the XPCS measurements (Fig. 1C). Our customized ruby fluorescence setup consists of a microscope (Navitar M5DC-KIT-CYL 5 MP), a laser (Roithner CW532-100, 532 nm) and a spectrometer (Ocean Optics HR4000). The position of the ruby fluorescence line shifts with pressure and temperature. The method is commonly used in diamond anvil experiments and well calibrated and documented in the literature[38–40]. We measured the position of the ruby R1 line at 93 K to be 693.360 ± 0.004 nm for sample A and <693.366 ± 0.003>nm for sample B (see Fig. S1-B). The shift of the ruby line is very small at the low pressures investigated here, a detailed discussion can be found in SI. We admit that this leads to a quite large error bar. Future experiments might consider to use an alternative pressure gauge[3]. The pressure, however, is also estimated from the X-ray scattering measurements (see details provided in SI). Given the uncertainty in temperature and peak position, we determined the pressure of both samples to be around 0.08 ± 0.02 GPa. The fact that both samples exhibit the same (or very similar) pressure is also reflected by the identical position of the first diffraction maximum (Fig. S2). Also note that the pressure is measured at the lowest temperature. During heating inside the DAC the P-T pathway will not be isobaric. We expect a small decrease in pressure (~11% in the range 93 K–123 K, see Fig. S1-C) within the investigated temperature range. Since the given pressure anyway carries a large error bar, we relate to the initial pressure throughout the manuscript. In-situ pressure determination during the XPCS experiments was not possible for technical reasons (see SI).

### X-ray photon correlation spectroscopy (XPCS)
We performed XPCS measurements in SAXS geometry at beamline P10 at PETRA III of the Deutsches Elektronen-Synchrotron (DESY) in Hamburg, Germany. We recorded SAXS patterns using an EIGER 4 M detector at a

sample-detector distance of 5 m, a photon energy of 12 keV, and a focused X-ray beam of 14.6 μm × 8.5 μm (horizontal × vertical). Usually, a series of 1000 images with 1 s exposure each was taken. As attenuators we used different stacks of polished silicon wafers adding up to a thickness of 300 μm or 450 μm of silicon, to mitigate radiation damage. This corresponds to an X-ray transmission of 0.27 and 0.14, respectively. Prior to each measurement the sample was equilibrated for 20 min. To monitor the molecular structure of the cold loaded sample, a second detector (EIGER 500 K) was mounted close to the chamber, covering a range of $1.4 \, \text{Å}^{-1} < Q < 2.9 \, \text{Å}^{-1}$ (Fig. 1A). $LaB_6$ was used for calibration. However, the WAXS detector is tilted with respect to the beam path and we estimate the uncertainty in $Q$ to be $\pm 0.013 \, \text{Å}^{-1}$ for the WAXS measurements. Wide angle scattering (WAXS) and SAXS data were collected simultaneously. Pseudo-Kossel lines[32] which appear on the detector images (Fig. 1A), are masked before the XPCS data analyses is performed. Data are analyzed using a software available at beamline P10.

## Data availability
Raw data from the X-ray measurements are stored on the cluster at the DESY infrastructure and are available upon request. Processed data and selected raw data are available on ZENODO via https://zenodo.org/doi/10.5281/zenodo.13766786.

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

## Acknowledgements

We thank Mathias Hudl from Stockholm University for help with setting up our customized and very compact Ruby system, which can now be transported and implemented at the beamline. We acknowledge Deutsches Elektronen-Synchrotron DESY (Hamburg, Germany), a member of the Helmholtz Association HGF, for the provision of experimental facilities. Parts of this research were carried out at PETRA III, beamline P10. Beamtime was allocated for proposal I-20200286 EC and I-20220659. We also thank Hans-Peter Liermann (DESY) for helpful discussion, as well as practical and technical help with our DAC experiments. Thanks to Anna Pakhomonova (DESY) for introducing us to the handling of the DAC when we started this project, and Iris Schwark and Anita Ehnes from the PETRA III sample environment group. This research was supported in part through the Maxwell computational resources operated at DESY, Hamburg, Germany. K.A.-W. acknowledges funding by the Ragnar Söderbergs Stiftelse (Sweden) and Carl Zeiss Stiftung (Germany). R.T. and S.H. acknowledge Academy of Finland projects 295696 and 303914. B.M. has received funding from the European Research Council (ERC) under the European Union's Horizon 2020 research and innovation programme (Grant agreement No. 101002868). T.E., N.N.S., and R.P.C.B. acknowledges funding by the Centre for Molecular Water Science (CMWS) within Early Science Projects. We also acknowledge the scientific exchange and support of the CMWS. We thank Gerhard Grübel, Werner Steffen and Fivos Perakis for scientific discussions.

## Author contributions

Following CRediT contributor roles, authors contributed as follows. Conceptualization & Supervision: K.A.-W., H.L., and F.L.; Investigation (experiments): A.K., H.L., T.E., M.L.-P., B.M., M.F., N.K., K.H., R.T., R.P.C.B., C.G., N.N.S., F.D., F.W., M.S., F.L., and K.A.-W.; Resources & Software (X-ray part): R.P.C.B., F.D., S.H., F.W., M.S., and F.L.; Resources (sample preparation): A.K., H.L., T.E., B.M., A.R., and K.A.-W.; Data analysis and Visualization: A.K., H.L., and T.E.; Writing original draft: A.K., F.L., and K.A.-W.; All authors have edited the draft and given approval to the final version of the manuscript.

## Funding

## Competing interests

The authors declare no competing interests.
