## [Transparent Peer Review file · Communications Chemistry]

Figures have been redacted from this file by request of the authors.

Multicomponent dynamics in amorphous ice studied using XPCS at elevated pressure and cryogenic temperatures

Corresponding Author: Professor Katrin Amann-Winkel

Version 0:

Reviewer comments:

Reviewer #1

(Remarks to the Author)

Water can be vitrified into a low-density amorphous ice (LDA) at low pressures, and a high-density amorphous ice (HDA) at high pressures (at approx. $P < 1$ GPa). LDA and HDA are remarkably different; in particular, they have different glass transition temperatures [$T_{g,LDA}(P)$ and $T_{g,HDA}(P)$]. While the $T_{g,HDA}(P)$ increases with increasing pressure, the $T_{g,LDA}(P)$ is expected to decrease with increasing pressure. Yet, the experimental data for $T_{g,LDA}(P)$ and $T_{g,HDA}(P)$ is scarce. In the case of HDA, measurements of $T_{g,HDA}(P)$ under pressure have been obtained in a handful of experiments, all based on very different experimental techniques (eg, NMR, calorimetry, dilatometry, etc). At high pressures, where experiments are difficult to perform, the values of $T_{g,HDA}(P)$ are rather scattered. In this study, the authors show that (i) it is possible to perform x-ray photon correlation spectroscopy (XPCS) experiments using a diamond anvil cell (DAC) to study the glass transition of water (HDA) under pressure. (ii) The T_g of HDA is measured at 0.08 GPa and it is shown that $T_{g,HDA}(P)$ increases with increasing pressure.

Technically, the experiments in this work are important since they introduce a novel tool (XPCS+DAC) to study the dynamics/glass transition of glassy water under pressure, and at cryogenic temperatures. From a fundamental point of view, this work provides a direct/in-situ measurement of the glass transition of HDA at elevated pressures (0.08 GPa), where experimental data is scarce. The results of this work are important to improve our understanding of the complex phase behavior of supercooled and glassy water. Accordingly, I find the manuscript suitable for publication in Comm. Chem. Below are a few comments that the authors should consider.

1) The glass transition of HDA should indicate the transformation of HDA to the liquid state, in principle, high-density liquid water (HDL). However, the transformation reported here (samples A and B) is from HDA to LDA. This should be clarified in the text. I believe that the authors refer to the "glass transition of HDA" as the temperature at which HDA transforms to either low-density liquid water (LDL), LDA, or HDL upon isobaric heating.

- One definition of T_g is given by the T at which $\tau \sim 100$ s. In Fig. 2D, $\tau > 100$ s at $T > 111$ K where the system is in the LDA state. Is this why it is stated that HDA transforms to LDA (instead of LDL)? Is there any other reason?

- If the effective heating rate in the experiments was slower, would the authors expect the sequence of transformations (upon heating) HDA-to-HDL-to-LDA, as found in previous studies at 1 bar? Or, HDA-to-HDL-to-LDL-to-LDA? On page 10, there is a discussion on the nature of the transformation of HDA upon heating but the discussion is very brief and condensed. It may be useful to expand it and make it clear what the potential interpretations of these experiments are [eg, HDA-to-LDA, HDA-to-(HDL+LDA+Ice), etc].

2) I wonder if the results in Fig. 7 for $T_{g,HDA}$ at $P=0.08$ GPa have any implication on the value of $T_{g,LDA}(P)$ at $P=0.08$ GPa. Since HDA transforms to LDA (instead of LDL) upon heating at $T_{g,HDA} \sim 130$ K, can one conclude that the $T_{g,LDA}(P=0.08 \text{ GPa}) > 130$ K? A discussion may be useful.

3) It would be useful if the "temperature" in the discussion and figs refers solely to the sample temperature, T_{sample} (as opposed to the cryostat temperature, T_{cryo}). For example, different temperatures are used in Figs. 7 and 2D and in the text.

4) Fig 7 is particularly important. It may be useful to use different symbols to distinguish the HDA-to-LDA and HDA-to-ice transformations observed in the 5 samples studied. Could the authors compare their results with available data for the $T_{g,HDA}(P)$ available in the literature (isobaric heating, calorimetry/volumetry/etc)?

Minor points:

- The abstract may need to be revised. E.g., the sentence "While...crystallization" is confusing.
- on page 2, it is stated that "...minimum in the Ih-III-liquid triple point at 2 GPa...". Should it be 0.2 GPa?
- there are no labels in the y-axis of Figs. 2C, 5C (compare with, eg, Fig. 4F).
- I found Table S1 and the middle/right panels of Fig. S1 to be very useful. They summarize nicely why the authors focus on samples A and B in the early sections of the main manuscript. As it is, this rationale is unclear (page 4, first parag). It may be good to include Table S1 and the middle/right panels of Fig. S1 in the main manuscript (as a new Fig 1), and add 2-3 lines in the first parag of page 4.
- On page 6, line 5, it is stated that "while Fig. 2A shows a clear LDA signal at this temperature". This is confusing. Should it be "HDA"?

Reviewer #2

(Remarks to the Author)

The manuscript reports a study on dynamics of a H₂O-eHDA sample in a diamond anvil cell (DAC) using synchrotron XPCS measurements by a strong team in this field. Of the five experimental runs, successful XPCS data collection is reported in two runs (A and B). Considering the experiment's challenges, the demonstrated feasibility of XPCS measurements on a small H₂O sample in a DAC is publishable and will interest researchers in glassy state and high-pressure research, as well as in condensed matter physics more broadly. The reported XPCS data show complex dynamics, requiring "multicomponent" to explain the experimental data. Overall, this study explores a new technique to address a significant problem. The XPCS data seem to be of good quality. One major concern of the study is the lack of in situ determinations of the pressure and temperature conditions where the XPCS and/or WAX measurements were made, which can lead to large systematic errors in the reported pressure and temperature values (see the comments below). Without proper assigned pressure-temperature conditions, it becomes difficult to understand the measured data. Consequently, the conclusions of the pressure dependencies of the glass transition temperature and the crystallization temperature are not justified. The conclusion on the effect of photon flux is also questionable. The manuscript requires major revision before being considered for publication, by focusing more on the experimental feasibility and spelling out the challenging aspects and the related efforts. The following comments may be considered in the revision.

1. Pressure determination:

A pressure of 0.08 GPa is reported for both runs of A and B using the ruby fluorescence technique. For such a small pressure value using a ruby scale, a reference at ambient pressure must be used for properly measuring the relative shift of the ruby fluorescence line. Without a proper reference at the same temperature, the reported pressure can carry unknown systematic errors (spectrometer calibration, temperature correction of the ruby shift, etc), making the reported 0.08 GPa a meaningless value.

The pressure was measured by ruby fluorescence only at the initial point at ~82 K after sample was loaded in a DAC. The DAC (and the samples) was then heated and XPCS was recorded at several temperatures, assuming no pressure change in the heating process. However, it is well known that the P-T pathway is not isobaric in a screw-driven DAC in a cryostat, with a potential pressure change more than the entire pressure range of this study. It is therefore important to monitor pressures as temperature is changed, because of the thermal effect on the cell body, the screw loading etc.

Diffraction peaks were also used for estimating pressures of the sample chamber in the runs of F and G (Table S1). In fact, this is a valid method, as it in situ monitors the pressures. Additional details should be provided regarding the XRD peak positions and the specific references used for pressure calculation. It is suggested that XRD information is used to estimate pressures for other runs as well (A,B,D). For example, the sample A at 82 K displays a shift in FSDP with time (Fig. S4), which means that there involves pressure change during the period.

2. Temperature calibration:

The offset in temperature between the cold finger and the sample chamber is based on the Fig. S3. However, there lack details of procedures for obtaining the data in Fig. S3. Because the offset is clearly dependent on thermal pathways (cooling, heating, and their rates), it is not a fixed number, but should vary in a range from positive to negative depending on the pathways and the experimental setup. Thus, a simple correction of a fixed 15 K based on Fig. S3 does not represent the actual temperatures of the sample.

3. A second sample spot within sample A:

It is unclear how the experiment for the second sample spot was conducted. Was the DAC cooled down back to 82 K and then heated again? If not, how was the initial time determined for XPCS, compared to the initial time of the first spot?

4. Effect of photon flux on dynamics:

The interpretation of the difference in XPCS results between runs A and B as due to photon flux on dynamics. As mentioned in the above comments, since the pressure conditions are uncertain in runs A and B, this interpretation is not conclusive. The effect of flux is only one of the factors among others such as different pressure (and temperature) conditions, different pathways (and durations) of the two samples.

5. Temperature intervals:

Most experimental data are presented at specific temperatures (e.g., 104 K, 111 K) as a function of time. What were the experimental procedures? What were the rates when the temperatures were raised from one level to the next? How were the initial times (t₀) determined? What were the uncertainties in temperature at a given level? It is difficult to understand the data without such important details.

6. Fig. S4

Figure S4 clearly shows that the FSDP position of HDA changes over time, even at 82 K, which contradicts the description in the main text.

Reviewer #3

(Remarks to the Author)
Dear Editor and authors,

you can find my report in attachment

Version 1:

Reviewer comments:

Reviewer #1

(Remarks to the Author)

The authors have addressed in detail the points raised in my previous report. I appreciate the clarification on the kinds of transformations observed upon heating (eg, HDA LDA, HDA HDL LDA/LDL) in page 10 (Conclusions), and the new Fig. 7. I find the new version of the manuscript suitable for publication in Communications Chemistry.

Reviewer #2

(Remarks to the Author)

Although the revision has shown some improvements, I find some of the responses to my previous comments not satisfactory, as listed below.

Pressure measurement: The authors seem to insist the validity of a pressure of 0.08 GPa by ruby fluorescence. Note that this 0.08 GPa corresponds to a ruby line shift of only ~ 0.003 nm, which is comparable to (even a little smaller than) the uncertainties as shown in the Fig. 1C, where the errors (from fitting statistics alone) in their fluorescence measurements are listed as ± 0.003 nm and ± 0.004 nm. This means that the employed ruby technique (and the spectrometer Ocean Optics HR4000) does not provide necessary precision to reliably measure a pressure as small as 0.08 GPa. In addition, I am troubled by the added error bar of ± 0.02 GPa in the revision, because the authors do not provide any basis of this estimate. Furthermore, when discussing the pressure drift as the cell temperature was changed, a 10% decrease in pressure was given/added in the revision, again without any support from measured numbers. The authors argue that the ruby fluorescence "was the best choice". But that choice does not justify the reported pressure values. In fact, the use of fluorescence of f-electron metals, e.g., $\text{Sm}^{2+}:\text{SrFCl}$ (<https://doi.org/10.1080/08957959.2021.1931168>) provides >3 times better resolution at low pressure.

On the other hand, x-ray data (FSDP, WAXS, transition) may give better precision in pressure determination in this study. The authors state that "Note, we do not use the crystalline Bragg peaks for calibration. The equation of states for hexagonal ice indicates that a peak shift in the relevant pressure range is very small ($< 0.005 \text{ \AA}^{-1}$) and therefore not detectable with our WAXS detector arrangement". This argument is invalid. To some degree, this argument applies better to the ruby technique in this case. In proper WAXS setups, such as the one reported in this study, a detectable limit should be better than 0.001 \AA^{-1} . The mentioned 0.005 \AA^{-1} range of peak shift not only should be detectable, but can be used for pressure determination. Generally speaking, no fluorescence technique can provide better precision in pressure determination than the x-ray diffraction technique.

Temperature offset: The response of the temperature offset is also not satisfactory. My comment was "Because the offset is clearly dependent on thermal pathways (cooling, heating, and their rates), it is not a fixed number, but should vary in a range from positive to negative depending on the pathways and the experimental setup". The authors only responded the effect of cooling rate (while their experiments were conducted under heating cycles). The authors mentioned (correctly) that "the thermocouple (sample temperature) lags behind", which means that the sample temperature could be higher than the cryostat temperature during cooling, and that the opposite could occur during heating cycle. That is the sample temperature could be lower than the cryostat temperature. If this is indeed the case, then the temperature correction should be opposite. The interpretation in the main text regarding the glass transition temperature should then be completely revised.

Reviewer #3

(Remarks to the Author)

I thank the authors for the effort they made to reply to all comments and improve the quality of their work. However, at the light of the responses to all referees and the new version of the manuscript, I am still not convinced on the data interpretation and accuracy. This is an important article that can pave the way to numerous experimental and numerical studies. I recognize the investment of the authors and their expertise in the field of amorphous water. Still, the argumentation employed by the authors to determine the glass transition temperature is not convincing in my opinion. I cannot therefore support the publication of the article. The reproducibility of the data of sample B in a second experiment, together with a better control on the pressure, would definitely help elucidating the origin of the observed dynamics.

- Scattering Pattern: I still believe that the speckle pattern in Fig. S2C has an additional contribution beyond the Kossel line, which arises from a second beam reflected by the diamonds through the sample. The Kossel line cannot explain the

additional speckles in the 3rd and 4th modules with a circular pattern not centred with respect to the direct beam (and a shape similar to the scattering of the sample). As the article does not mention whether this scan corresponds to sample A or B, it is impossible to know the effect on the dynamics. I suggest the authors to verify that all scans used to calculate the dynamics are clean, with only Kossel lines (as in the new Fig. 1 where the authors did the azimuthal test).

- Glass transition identification: I still don't understand how the induced dynamics of sample A which is influenced by a structural damage can be used to strengthen the validity of the measured dynamics of sample B. Furthermore, even if the motion is more diffusive in sample B at high temperature, the strength of this process is reduced, indicating that this process is less dominant, so I don't see how this can be attributed to the glass transition, and why this aspect is not commented in the text. As written in my previous report, I would appreciate also if the authors can provide an evidence of the fact that the dynamics at T_g should become diffusive. There are different theoretical works in glass formers suggesting a completely different mechanism of particle motion in supercooled liquids (see for instance Bhattacharyya, J. Chem. Phys. 132 104503, 2010). This is still not commented. Furthermore, at 136K the "The WAXS image at this temperature shows a small amount of crystalline ice, but no drastic growth". It is well known that dynamic probes are more sensitive than single point correlators. How can the authors exclude that crystallization already affect the dynamics at 130K? This would indeed explain why the strength of the diffusion process decreases.

- "With which criterion the temporal binning for the correlation functions are selected?" With this question I was meaning the number of frames used to get correlation curves. This is completely arbitrary and not enough commented in the text. As the TTC are not stationary, this choice can dramatically affect the results.

- I am still confused by the pressure determination and stability. There is an uncertainty of 10% as pressure is measured only at the beginning without considering temperature effects. Then there is an uncertainty due to the fact that pressure is checked by the ruby luminescence spectra which is challenging for low pressure values. Finally there is an uncertainty due to the fact that pressure is calibrated from the evolution of the amorphous structure measured in a set-up which is not optimized for diffraction (and without using a crystalline calibrant). At the light of all this, how is possible to have "The error in pressure determination is at least ± 0.02 GPa"? This value seems to low for this set-up. How this is estimated? More importantly, pressure should remain constant during the measurements. How can the authors be sure about it?

To conclude, I believe that this work is very interesting and important but that it is still far from clear enough to be published.

Version 2:

Reviewer comments:

Reviewer #2

(Remarks to the Author)

In my previous review, I mistakenly put a shift of 0.003 nm (instead of 0.03 nm) for a pressure of 0.08 GPa. I thank the authors for clarifying this and the added information in the revision. I also find the responses to the other comments reasonable. In my opinion, the revised manuscript is ready for publication with minor modification (see below).

On the temperature offset, it appears that there is always a large temperature difference between the thermocouple readings and diode readings, even after a few hours (see figs. S3-D and S4). Initially I assumed that these two temperatures (from thermocouple and diode) suppose to be close within a few degrees after equilibrium is reached. Apparently their measured results before and in more recent repeated experiments show that thermocouple readings are consistently at higher temperatures. This could be related to the cryostat design, the sensor calibration, the heater location, etc. A note on this would help readers to understand the positive offset corrections in the heating cycles.

Reviewer #3

(Remarks to the Author)

I thank the authors for their detailed answer. They have addressed all the points raised in my previous report in an accurate way. I therefore support the publication of this article in Communications Chemistry. I am sure that it will pave the way to new interesting studies.

We would like to thank the reviewers for their hard work and valuable comments. We appreciate the time and effort they have dedicated to this review. Below is a point-to-point response to the different comments:

Reviewer #1:

Water can be vitrified into a low-density amorphous ice (LDA) at low pressures, and a high-density amorphous ice (HDA) at high pressures (at approx. $P < 1 \text{ GPa}$). LDA and HDA are remarkably different; in particular, they have different glass transition temperatures [$T_{g,LDA(P)}$ and $T_{g,HDA(P)}$]. While the $T_{g,HDA(P)}$ increases with increasing pressure, the $T_{g,LDA(P)}$ is expected to decrease with increasing pressure. Yet, the experimental data for $T_{g,LDA(P)}$ and $T_{g,HDA(P)}$ is scarce. In the case of HDA, measurements of $T_{g,HDA(P)}$ under pressure have been obtained in a handful of experiments, all based on very different experimental techniques (eg, NMR, calorimetry, dilatometry, etc). At high pressures, where experiments are difficult to perform, the values of $T_{g,HDA(P)}$ are rather scattered. In this study, the authors show that (i) it is possible to perform x-ray photon correlation spectroscopy (XPCS) experiments using a diamond anvil cell (DAC) to study the glass transition of water (HDA) under pressure. (ii) The T_g of HDA is measured at 0.08 GPa and it is shown that $T_{g,HDA(P)}$ increases with increasing pressure.

Technically, the experiments in this work are important since they introduce a novel tool (XPCS+DAC) to study the dynamics/glass transition of glassy water under pressure, and at cryogenic temperatures. From a fundamental point of view, this work provides a direct/in-situ measurement of the glass transition of HDA at elevated pressures (0.08 GPa), where experimental data is scarce. The results of this work are important to improve our understanding of the complex phase behavior of supercooled and glassy water. Accordingly, I find the manuscript suitable for publication in Comm. Chem. Below are a few comments that the authors should consider.

We thank the reviewer for supporting the publication of our manuscript. Below we address the raised questions.

1) The glass transition of HDA should indicate the transformation of HDA to the liquid state, in principle, high-density liquid water (HDL). However, the transformation reported here (samples A and B) is from HDA to LDA. This should be clarified in the text. I believe that the authors refer to the “glass transition of HDA” as the temperature at which HDA transforms to either low-density liquid water (LDL), LDA, or HDL upon isobaric heating.

- One definition of T_g is given by the T at which $\tau \sim 100\text{s}$. In Fig. 2D, $\tau > 100\text{s}$ at $T > 111\text{K}$ where the system is in the LDA state. Is this why it is stated that HDA transforms to LDA (instead of LDL)? Is there any other reason?

This point indeed needs some clarification. As onset of the glass transition T_g we refer to a transition from HDA to HDL. We identify this by the appearance of a diffusive component. At high enough temperatures the transition is expected to take the path HDL→LDL. Please also see our answer to your next question below.

In literature, T_g is defined as the temperature where the relaxation time is 100 s. However, this refers to molecular length scales, hence the self-diffusion. In the current setup we are probing in small-angle geometry (SAXS), and therefore probe larger length-scales. This is why the extracted relaxation time τ of 100 s does not directly relate to T_g .

To make these points clear, we added the following in the manuscript on page 4:

Please note, that the here derived relaxation time is measured in SAXS geometry at a momentum transfer range of $0.002 \text{ \AA}^{-1} < Q < 0.02 \text{ \AA}^{-1}$ and hence probes a collective motion at several 100 nm length scales rather than the molecular self-diffusion. A relaxation time of 100 s therefore does not automatically relate to the glass transition, as known e.g. from dielectric measurements.

- If the effective heating rate in the experiments was slower, would the authors expect the sequence of transformations (upon heating) HDA-to-HDL-to-LDA, as found in previous studies at 1 bar? Or, HDA-to-HDL-to-LDL-to-LDA? On page 10, there is a discussion on the nature of the transformation of HDA upon heating but the discussion is very brief and condensed. It may be useful to expand it and make it clear what the potential interpretations of these experiments are [eg, HDA-to-LDA, HDA-to-(HDL+LDA+Ice), etc].

As suggested by the referee, we extended the discussion on page 8 and 10 as follows:

Page 8:

...the related TTC shows a deceleration over time (Fig. S10). Figure S10 in addition shows the XPCS analysis after the transition to LDL or LDA and ice I. At 130 K the data let us assume a liquid-like character of the sample (LDL). A separate measurement on pure LDA over the whole temperature range would be necessary to confirm the location of the glass transition in LDA and the here observed liquid-liquid transition free of doubt. A detailed discussion of the LDL state can be found in the SI. We here focus on the analysis of the diffusive process related to the formation of HDL.

Page 10:

Based on this finding, T_g at $\sim 0.08 \text{ GPa}$ was estimated to be at 124 K (black spot, Fig. 7). To summarize, sample A follows a direct (presumably beam driven) transition $\text{HDA} \rightarrow \text{LDA}$, while sample B undergoes the pathway $\text{HDA} \rightarrow \text{HDL} \rightarrow \text{LDL/LDA}$. Literature as well as the here presented data, let us assume, that sample B proceeds a liquid-liquid transition ($\text{HDL} \rightarrow \text{LDL}$) at this pressure and temperature range. Our data demonstrate that the glass transition temperature T_g for HDA, as measured by XPCS, has a positive pressure dependence.

In addition, we added the previously not shown XPCS analysis of the higher temperatures of sample B to SI (Fig. S10), including discussion.

Figure S10 in addition shows the XPCS analysis after the transition to LDL or LDA and ice I. The question at hand is, which of the processes dominates, as it is very difficult to distinguish. A separate measurement on pure LDA over the whole temperature range would be necessary to confirm a potential liquid-liquid transition free of doubt.

At 130 K and 135 K the data let us assume a liquid-like character of the sample (LDL), similar to the previous temperature, based on the fits shown in Fig. S10. The dataset right after the transition shows a Q^2 -dependence and a diffusion coefficient of $D = 36.4 \text{ \AA}^2/\text{s}$. The Kohlrausch–Williams–Watts (KWW) exponent of the correlation decay approaches 1, which hints towards a Brownian motion. Still, the Q -dependency of the relaxation time τ exhibits an offset, and therefore no fully Brownian motion is observed. At higher temperatures a deceleration can be observed in the TTCs, this behavior is clearly opposite to the transition from high-to low-density. At 136 K this is accompanied by a second dynamical process, and a pronounced double exponential decay is visible, both in the g_2 curves as well as in the TTC.

In addition, γ for the faster process clearly changes to be < 1 . The WAXS image at this temperature shows a small amount of crystalline ice, but no drastic growth. Therefore, the observed dynamics can be related to both, formation of LDL as well as crystalline ice.

Fig. S10: (Top row) Autocorrelation functions g_2 and the corresponding KWW fits of sample B, calculated for the first 500 s of a series at $Q = 0.0035 \text{ \AA}^{-1}$. (Middle row) The results for the KWW exponent of fitting the g_2 functions of sample B. (Middle row) Q-dependency of the relaxation time τ . (Lower row) TTC for the three measurements.

2) I wonder if the results in Fig. 7 for T_g, HDA at $P=0.08$ GPa have any implication on the value of $T_g, \text{LDA}(P)$ at $P=0.08$ GPa. Since HDA transforms to LDA (instead of LDL) upon heating at $T_g, \text{HDA} \sim 130$ K, can one conclude that the $T_g, \text{LDA}(P=0.08 \text{ GPa}) > 130$ K? A discussion may be useful.

From simulations in literature, we expect the glass transition of LDA to be nearly constant or showing a slightly negative pressure dependence. As discussed now in the modified text (see previous answer), and new figure S9, one would need to measure pure LDA over the whole temperature range to determine the glass transition temperatures. We added additional data on LDA/LDL, showing that the sample rather transforms to LDL and not to LDA, the text is modified accordingly.

3) It would be useful if the “temperature” in the discussion and figs refers solely to the sample temperature, T_{sample} (as opposed to the cryostat temperature, T_{cryo}). For example, different temperatures are used in Figs. 7 and 2D and in the text.

We now use the calibrated sample temperature for all figures in the manuscript, in order to improve the readability and make a comparison to literature easier. A detailed description to the temperature protocol is now documented in the SI (page 3-4).

While adding the protocol, we noticed that we did a mistake in the previous version of the manuscript in labelling the temperatures. As we now describe in more detail in table 2 (SI), the 15 K temperature offset was measured using Diode A as a reference, while the previous temperature T_{cryo} was measured by Diode B. We hope the new detailed description provides a better understanding of how the temperature was measured and calibrated, and apologise for the confusion.

4) Fig 7 is particularly important. It may be useful to use different symbols to distinguish the HDA-to-LDA and HDA-to-ice transformations observed in the 5 samples studied. Could the authors compare their results with available data for the $T_g, \text{HDA}(P)$ available in the literature (isobaric heating, calorimetry/volumetry/etc)?

We thank the reviewer for the suggestion. In combination with other adaption, the diagram changed now as follows. As suggested by referee 3, we removed samples G and F, due to large uncertainties in the pressure determination. As suggested by referee 1, we use different symbols for different transformations. This is, circles represent the glass-transition, squares show the transition to LDA/ice-I and the black triangle is used for the transition to ice IX. We also added literature values from volumetric T_g measurements on HDA (open circles Seidl et al., 2011, ref. 6), the calorimetric T_g (open star, ref. 4) as well as in-situ dielectric measurements (open diamond, ref. 7). Note, that the temperature values have changed compared to the previous version of the figure, as explained in our previous answer to your 3rd question (see SI for details). The conclusion (page 10) is adapted as follows:

This is consistent with other findings in literature, which determined the glass transition temperature from volumetric measurements using a piston cylinder setup⁶, dielectric spectroscopy⁷ and MD simulations⁸. The experimental data are compared directly within Fig. 7 (open circles and open diamond, respectively). The offset between the datasets, e.g. to

the volumetric data, is well-founded by the distinct measures, different pathways (isobaric vs. not), heating rate and corresponding crystallization temperatures. Despite the offset, all methods show the same trend, demonstrating that the glass transition temperature increases with pressure.

Figure 7. A schematic phase diagram showing the transition- and crystallization temperatures of *three* eHDA samples measured at different pressures. Sample A (red) and B (green) transform predominantly to LDL (and ice I). Sample D (black triangle) crystallizes directly to ice IX. From sample B, T_g was determined by XPCS at ~ 0.08 GPa and 124 K (black spot). Other literature values for the glass transition T_g are shown for comparison, determined by different experimental methods as cited in the references.

Minor points:

- The abstract may need to be revised. E.g., the sentence “While...crystallization” is confusing.

Abstract is modified as follows:

The experimental identification of the glass transition at elevated pressure and cryo-conditions is technically difficult. Moreover, in the case of amorphous ices, the glass transition is interrupted by crystallization, which makes it even more challenging.

- on page 2, it is stated that “...minimum in the Ih-III-liquid triple point at 2 GPa...”. Should it be 0.2 GPa?

We are sorry for the typo and thank the referee for spotting this. Text is changed accordingly.

- there are no labels in the y-axis of Figs. 2C, 5C (compare with, eg, Fig. 4F).

We changed the figure and added the labels.

- I found Table S1 and the middle/right panels of Fig. S1 to be very useful. They summarize nicely why the authors focus on samples A and B in the early sections of the main manuscript. As it is, this rationale is unclear (page 4, first parag). It may be good to include Table S1 and the middle/right panels of Fig. S1 in the main manuscript (as a new Fig 1), and add 2-3 lines in the first parag of page 4.

We thank the reviewer for this suggestion. However, since we removed samples G and F from the main manuscript, we prefer to show the table in the supplementary information only.

- On page 6, line 5, it is stated that “while Fig. 2A shows a clear LDA signal at this temperature”. This is confusing. Should it be “HDA”?

Actually, no, it refers to the 111 K, yellow line in Fig. 2A. This I(Q) shows LDA.

We have now added the colour label in the text.

Reviewer #2:

The manuscript reports a study on dynamics of a H₂O-eHDA sample in a diamond anvil cell (DAC) using synchrotron XPCS measurements by a strong team in this field. Of the five experimental runs, successful XPCS data collection is reported in two runs (A and B). Considering the experiment's challenges, the demonstrated feasibility of XPCS measurements on a small H₂O sample in a DAC is publishable and will interest researchers in glassy state and high-pressure research, as well as in condensed matter physics more broadly. The reported XPCS data show complex dynamics, requiring "multicomponent" to explain the experimental data. Overall, this study explores a new technique to address a significant problem. The XPCS data seem to be of good quality. One major concern of the study is the lack of in situ determinations of the pressure and temperature conditions where the XPCS and/or WAX measurements were made, which can lead to large systematic errors in the reported pressure and temperature values (see the comments below). Without proper assigned pressure-temperature conditions, it becomes difficult to understand the measured data. Consequently, the conclusions of the pressure dependencies of the glass transition temperature and the crystallization temperature are not justified. The conclusion on the effect of photon flux is also questionable. The manuscript requires major revision before being considered for publication, by focusing more on the experimental feasibility and spelling out the challenging aspects and the related efforts. The following comments may be considered in the revision.

1. Pressure determination:

A pressure of 0.08 GPa is reported for both runs of A and B using the ruby fluorescence technique. For such a small pressure value using a ruby scale, a reference at ambient pressure must be used for properly measuring the relative shift of the ruby fluorescence line. Without a proper reference at the same temperature, the reported pressure can carry unknown systematic errors (spectrometer calibration, temperature correction of the ruby shift, etc), making the reported 0.08 GPa a meaningless value.

We agree that the ruby fluorescence technique at such low pressures is far from optimal. However, for the presented experiment this was the best choice we could make. We added a more detailed description to avoid misunderstandings. However, we did indeed use the Ruby reference at room temperature back then. Please also note, that all pressures not solely rely on the ruby measurement, but are backed up by the X-ray data and the location of the HDA halo peak as well as the transition to crystalline ice. This is explained further in our reply to the next question (see below). Note here, we added the error bars to all mentioned pressures and the following text:

Page 3 (introduction):

The error in pressure determination is at least ± 0.02 GPa, due to limitations of the used method.

Page 12:

The shift of the ruby line is very small at the low pressures investigated here, although the spectrometer is able to resolve the small shift. We admit that this leads to a quite large error bar. The pressure, however, is also estimated from the X-ray scattering measurements (see details provided in SI).

SI, page 2 (full paragraph also copied here to page 8, as reply to next part of the question):

Additional evidence for the sample pressure ...

The pressure was measured by ruby fluorescence only at the initial point at ~82 K after sample was loaded in a DAC. The DAC (and the samples) was then heated and XPCS was recorded at several temperatures, assuming no pressure change in the heating process. However, it is well known that the P-T pathway is not isobaric in a screw-driven DAC in a cryostat, with a potential pressure change more than the entire pressure range of this study. It is therefore important to monitor pressures as temperature is changed, because of the thermal effect on the cell body, the screw loading etc.

We thank the referee for pointing this out, as we forgot to mention this in the previous version of the manuscript and added some sentences for clarification to the text.

We are aware of the fact that the pressure inside the DAC changes with temperature.

We measured the pressure change at different pressures throughout the whole temperature range up to room temperature. The pressure in the temperature range 80 - 130 K, which is where the experiments are conducted, has a rather small variation of ~10%.

We added the following sentence on page 13:

Also note that the pressure is measured at the lowest temperature. During heating inside the DAC the P-T pathway will not be isobaric. We expect a small decrease in pressure (~10%) within the investigated temperature range. Since the given pressure anyway carries a large error bar, we relate to the initial pressure throughout the manuscript. In-situ pressure determination during the XPCS experiments was not possible for technical reasons (see SI).

Diffraction peaks were also used for estimating pressures of the sample chamber in the runs of F and G (Table S1). In fact, this is a valid method, as it in situ monitors the pressures. Additional details should be provided regarding the XRD peak positions and the specific references used for pressure calculation. It is suggested that XRD information is used to estimate pressures for other runs as well (A,B,D). For example, the sample A at 82 K displays a shift in FSDP with time (Fig. S4), which means that there involves pressure change during the period.

The pressure was estimated based on two methods.

- The position of the first diffraction maximum of the HDA.
- The observed phase transition at $T > 140$ K

Nelmes et al. (Nature Physics 2, 2006) reported on the Q-shift of the FSDP of eHDA (red symbols) inside a Paris Edinburg cell using neutron diffraction. Additionally, a density and d-spacing relation was reported on quench recovered samples, recovered from different pressures (Loerting JPCB 2011). Both figures from literature are shown here in our reply for reviewing purposes only (without permission by the publisher).

We observe a shift from $Q = 2.03 \text{ \AA}^{-1}$ (sample A/B) to $Q = 2.17 \text{ \AA}^{-1}$ (sample D) for samples measured at the lowest temperature. This clearly shows an increase in pressure, as always the same initial sample was loaded to the DAC and subsequently compressed. The position of sample D is in very good agreement with a pressure of 0.75 GPa, as determined by the Ruby setup. The pattern of samples A/B are almost identical to the quench recovered samples prior loading.

Figure-review-only: Literature data, Nelmes, Nat. Phys. 2, 2006 and Loerting JPCB 2011. Right figures show data from this study, now also added to figure S1.

We can also estimate the pressure from the observed phase transition at $T > 140$ K. This is, as summarized in Figures 7 and S1, sample A and B transform to a mixture of LDA and ice I, hence need to be located at low pressures in the phase diagram. Sample D transforms to ice IX, this is, the sample needs to be located at $p > 0.1$ GPa. This estimation is based on the paper by Seidl et al. (PRB 88, 2013). For reviewing purposes, we have added the relevant figures here. Phase diagram on the right is taken from Amann-Winkel et al., Rev. Mod. Phys 88, 2016:

Figure-review-only: Literature data, Seidl PRB 88, 2013 and Amann-Winkel Rev. Mod. Phys 88, 2016.

We added this discussion into the SI (page 2) and show the WAXS comparison in figure S1.

Additional evidence for the sample pressure is the location of the first broad diffraction maximum as well as the observation to which phase the eHDA sample transforms to upon heating. [3], [4] The shift of the Q_{max} of eHDA upon pressure is well known from literature. [5] Additionally, a density and d -spacing relation was reported on quench recovered samples, recovered from different pressures. [6] We here observe a shift from $Q = 2.03 \text{ \AA}^{-1}$ (sample A/B) to $Q = 2.17 \text{ \AA}^{-1}$ (sample D). This clearly shows an increase in pressure, as always the same initial sample was loaded to the DAC and subsequently compressed. The position of sample D is in very good agreement with a pressure of 0.75 GPa, as determined by the Ruby setup. The pattern of samples A/B are almost identical to the quench recovered samples prior loading. While samples A and B transform to LDA(L) before crystallization, indicating a pressure < 0.1 GPa, samples D directly transform to ice IX, indicating a pressure > 0.1 GPa.

Note, we do not use the crystalline Bragg peaks for calibration. The equation of states for hexagonal ice indicates that a peak shift in the relevant pressure range is very small ($< 0.005 \text{ \AA}^{-1}$) and therefore not detectable with our WAXS detector arrangement.

2. Temperature calibration:

The offset in temperature between the cold finger and the sample chamber is based on the Fig. S3. However, there lack details of procedures for obtaining the data in Fig. S3. Because the offset is clearly dependent on thermal pathways (cooling, heating, and their rates), it is not a fixed number, but should vary in a range from positive to negative depending on the pathways and the experimental setup. Thus, a simple correction of a fixed 15 K based on Fig. S3 does not represent the actual temperatures of the sample.

The referee is correct, that heating and cooling rate would affect the actual temperature. In the SI only the direct cooling is shown. In addition, we performed a cooling measurement with a slower rate, as well as a stepwise heating. While the diode changes its temperature exactly at a given rate of 5 K/min, the thermocouple (sample temperature) lags behind. At each temperature we waited for 20-30 mins, then a constant offset of 15 K was established.

As mentioned in the method section, prior to each XPCS measurement the sample was equilibrated for at least 20 min. Therefore, we are confident that a constant offset of 15 K represents well the actual sample temperature. A detailed description to the temperature protocol is now documented in the SI.

However, while adding the protocol, we noticed that we did a mistake in the previous version of the manuscript in labelling the temperatures. The 15 K temperature offset was measured using Si-diode A as a reference, while the previous temperature T_{cryo} was measured by diode B. While diode A is located inside the cold finger, diode B was placed on the copper holder, but still far away from the DAC sample compartment. We hope the new detailed description in the SI provides a better understanding of how the temperature was measured and calibrated, and apologise for the confusion. In the main text we refer to SI as follows:

The temperature calibration is described in detail in the SI, see Fig. S2-D, Fig. S3 and table 2.

3. A second sample spot within sample A:

It is unclear how the experiment for the second sample spot was conducted. Was the DAC cooled down back to 82 K and then heated again? If not, how was the initial time determined for XPCS, compared to the initial time of the first spot?

We added the following diagram to the SI, which shows the thermal history of the two samples. The second spot was measured directly after the first, without cooling in between. The spots are separated by 150 μm (motor in y-direction). Further details are provided on page 3-4 of SI.

Fig. S3: *Temperature protocol for samples A and B. XPCS data shown in the main manuscript are marked colored and include the timestamp from during the experiment.*

4. Effect of photon flux on dynamics:

The interpretation of the difference in XPCS results between runs A and B as due to photon flux on dynamics. As mentioned in the above comments, since the pressure conditions are uncertain in runs A and B, this interpretation is not conclusive. The effect of flux is only one of the factors among others such as different pressure (and temperature) conditions, different pathways (and durations) of the two samples.

As explained above, we are confident that the pressure conditions are similar for both samples. A difference in pressure would not have such a drastic effect on thermal stability. We know from several experiments at ambient pressure, that a different waiting time at the lowest temperature (82 K) would not cause the sample to transform to LDA. The duration at the higher temperature is comparable in both cases (see Fig. S3). In Fig. S5 we present WAXS data during 1000 s which clearly shows that the sample starts to transform during the XPCS run. The signal is relatively stable during the first 300 s, followed by a clear change. In comparison, sample B is much more stable during the full XPCS run in Fig. S8, exposed with a lower dose rate. These observations support the idea of an X-ray beam induced heating, but we agree that we cannot completely rule out other factors.

We therefore added the following to the text on page 10

...although other factors like the thermal history, cannot be fully excluded.

5. Temperature intervals:

Most experimental data are presented at specific temperatures (e.g., 104 K, 111 K) as a function of time. What were the experimental procedures? What were the rates when the temperatures were raised from one level to the next? How were the initial times (t_0) determined? What were the uncertainties in temperature at a given level? It is difficult to understand the data without such important details.

A detailed temperature protocol is now presented in the SI, see also previous answer. We added the following text to SI in order to provide detailed information. We selected the temperatures in samples A and B which allowed us to compare the two samples as we follow the phase transition. The text in SI (page 4) reads as follows:

...During the experiments, the temperature at the bottom of the cryostat was measured using the inbuilt Si-diode (sensor A) as well as an additional Si-diode (sensor B, as visible in Fig. S2B₁). During heating, sensor A and B show a small thermal lag, sensor B follows sensor A with some delay, but stabilizes after 20 min. Figure S3 and Table S2 show the temperature protocol of samples A and B. The sample temperature inside the DAC must have an even larger offset. After the X-ray experiments, we calibrated this offset by gluing a thermocouple (type K) onto the gasket inside the DAC. The results for the cooling are shown in Fig. S2D, data during step-wise heating with 30min annealing time, similar to the experiment, are shown in Table S2. The temperature offset was determined to be 15 ± 2 K...

...The temperature during the experiment was changed at a rate of 5 K/min. The inbuilt Si-diode (sensor A), as purchased from JANIS, exactly follows at this rate, we did not observe any overshoot effect, once the set-temperature is reached the given value oscillates by ± 0.1 K (slightly more at lower temperatures). We changed the temperature in steps of 10 K at low temperature and 3-4 K at the higher temperatures. The exact protocol is graphically shown in Fig. S3. At each temperature step we measured three to four XPCS series (1000s each) at different sample spots, the series discussed within the paper are marked coloured. The first series was started directly when sensor A reached the set temperature, and is affected by equilibration. This is, it usually shows faster and heterogeneous dynamics, therefore only the series measured after the equilibration time of 20 min (1000s + time to find a new sample spot) or at later times is used for the analysis. Searching a new spot and changing temperature is done by hand and not fully automated.

6. Fig. S4

Figure S4 clearly shows that the FSDP position of HDA changes over time, even at 82 K, which contradicts the description in the main text.

As discussed above, for sample A the small change observed in the WAXS pattern might be caused by X-ray induced heating, even at the lowest temperature. In the main text we only refer to a stable WAXS pattern over a time interval of 300s. We added "300s" to that sentence in order to avoid confusion. In the figure below, the first 300s in 100s intervals are plotted separately for reviewing purposes.

Figure-review-only: Sample A, WAXS during the first 300 s.

Reviewer #3:

In XPCS at elevated pressure and cryogenic temperatures: Multicomponent dynamics in amorphous ice Karina and co-authors investigate the dynamics of amorphous ice at 0.08GPa during its transformation from a HDA to LDA. Water has a complex and debated phase diagram with many crystalline phases and at least two amorphous ice solids phases. These amorphous phases are fundamental in many cosmological processes and have been often connected to the well-known anomalies of liquid water. In this technical article, the authors present a challenging experiment to measure the glass transition of amorphous ice at elevated pressures and cryogenic temperatures from time-resolved speckle autocorrelation measurements of x-rays scattered by the sample. The authors claim to see the presence of two dynamical contributions, a ballistic and a diffusive one, during the HDA-LDA transition and they located the glass transition temperature of LDA. This is an extraordinary claim, however the evidence provided for it raises several crucial questions that I think the authors should address

- The SAXS profile presents an additional contribution which is discussed in terms of pseudo Kossel lines. It looks to me that alternative explanations could also be considered and discussed. A similar pattern could for instance arise from a multiple scattering of the sample scattering, superimposed on the first one, probably reflected by a diamond acting as an analyser. This could explain the partial scattered profile, centred at a different position which is cut by the misalignment of the diamond. A simple rotation of the cell should verify this hypothesis. The presence of a secondary SAXS pattern could explain the “interaction with the speckles patterns” even outside the “masked pixels”. Have the authors compared the analysis of the bottom part of the SAXS profile with the top part? Is it the same? This aspect is fundamental to assess the reliability of the results.

We understand the concerns raised by the referee. However, the sample is only 60 μm thick. At a photon energy of 12 keV and this sample thickness, multiple scattering as well as secondary scattering on the second diamond can be excluded. We actually did try to rotate the sample to get rid of the lines to no avail. Nevertheless, please note that within the analysis of samples A and B weak Kossel lines are present. In order to avoid confusion, we replaced the detector image in figure 1, as this was a detector image from sample D at a higher pressure. This detector image is now shown in the SI only in Figure S2-C.

We analysed the XPCS pattern at different parts of the detector and found them to be identical. This information and the figure is now provided in SI, Fig. S4:

...The image in Fig. S2C shows a strong line visible at a pressure of 0.75 GPa in the 2D SAXS pattern. These lines do not vanish when rotating the DAC inside the X-ray beam.

.... Sample D could therefore not be analyzed in terms of dynamics. For lower pressures as in samples A and B, weak Pseudo-Kossel lines can be masked out, without influencing the result. No effect of directionality has been found. Figure S4 shows the masked region on the detector, and g_2 curves calculated independent from the upper and lower part of the detector.

Error! Bookmark not defined. Fig. S4: Example for g_2 curves calculated separately from the upper and lower part of the detector.

- For what discussed in the work, is clear that Sample A suffers of radiation damage: the transition is at the wrong temperature, wrong dynamics, already at 82K in figure S4 the WAXS structure is not stable and changes with time, and even the SAXS $I(Q)$ has a different profile in the two samples. After the transition, the $I(Q)$ of sample A is basically a straight line in log scale. So, as the authors as well are considering sample A not reliable, I don't understand why the measured dynamics is still discussed and considered partially representative of the transition HDA- LDA. It could be directly moved to SI. the data of sample A are all measured on a same spot and at 104K on a second position. As the beam is locally heating or damaging, the two datasets cannot be compared having different doses.

The sample not only transforms at too low temperature but also shows clearly distinct dynamics, namely relaxation times with a linear Q -dependence, which indicate a directed motion rather than Brownian motion. This is actually why we believe that the dataset is important to be shown in the main manuscript as well. It is important to highlight the difference in the dynamic behaviour and supports the finding in sample B, namely liquid-like dynamics appearing in sample B.

- Data analysis lacks of some information. How are the baselines determined? Is the contrast constant during the transition?

We used a calibration measurement of Aerogel at the beginning of the experiment, to establish the maximum contrast. We now also added more detailed information in SI including contrast and baseline parameters. The following to the main text on page 9:

... Note that we did not – as is commonly done – omit the cross-term that arises when expanding the squared expression. Additional information on this model, the fitting procedure and their parameters is available in SI.

Updated SI on page 9:

Simultaneously fitting of g_2 curves with two dynamical components

In the main manuscript we discussed that the correlation functions of sample B (Fig. 5) can be best modelled by simultaneously fitting two dynamical components (see Fig. 6). The model in equation (5) is a linear combination of a ballistic and a diffusive component, with v as velocity and D as diffusion coefficient:

$$g_2(Q, \Delta t) = b(Q) + \beta(Q) \cdot \left(A e^{-(vQ\Delta t)^2} + (1 - A) e^{-(DQ^2\Delta t)^\gamma} \right)^2.$$

The parameters are as follows: baselines $b(Q)$, contrasts $\beta(Q)$, ballistic amplitude A , diffusive stretch/KWW exponent γ , ballistic velocity v , and diffusivity D . The model is adapted to all data points $g_2(Q, \Delta t)$ in a measurement series with a least-squares surface fit. For optimization, we used the `curve_fit` function from SciPy's optimize subpackage [8] (with the default trust region reflective minimization algorithm). There is one baseline parameter b and one contrast parameter β per Q -value, so in total, $2 \cdot N_Q + 5$ parameters are fitted to each measurement (N_Q is the number of Q bins).

We used a calibration measurement of an Aerogel sample to establish the maximum contrast, here determined to be $\sim 9\%$. Baselines which are affected by e.g. streaks or a Kossel-line, given by the difference of 1 and the g_2 value at $t \rightarrow 0$, are corrected by the aerogel contrast. The fit results for contrast parameter β and baseline parameter b , related to Fig. 6 of the main manuscript, are shown in Fig. S11. The contrast drop with increasing momentum transfer Q matches our expectations.

Fig. S11: Contrast β and baseline values b determined by fit to the autocorrelation functions g_2 of sample B. This figure relates to Fig. 6 of the main manuscript, values shown for temperatures $T_{\text{sample}} = 118 \text{ K}, 121 \text{ K}, 124 \text{ K}$.

With which criterion the temporal binning for the correlation functions are selected?

The g_2 functions were calculated using a software available at P10. It is based on a multi-tau algorithm that naturally results in an almost logarithmic binning of the delta t values.

This information is added to the main manuscript on page 4:

The software is based on a multi-tau algorithm that naturally results in an almost logarithmic binning of the Δt values.

And page 13 (materials and methods): *Data are analyzed using a software available at beamline P10.*

Looking at figure 6, not all data sets are faster after 500s.

Indeed we only observe a clear acceleration in the TTCs for the two highest temperatures, while the TTC at 82 K (now given as a sample temperature of 93 K, see T-calibration) is rather stable throughout the 1000 s of the measurement, reflecting the arrested glassy state. In fact, the fit results for $T = 82$ K using 1000 s are almost identical. Nevertheless, we wanted to fit a comparable time range for all temperatures and therefore decided to analyse 500s consequently for all measurements.

Although the authors claimed the opposite, the strength of the diffusive process decreases at higher temperature (A is maximum at 115K). How can this process be dominant if its strength is so low?

We are sorry for the misunderstanding. When we wrote “we observe that diffusivity rises sharply between 121 K (111 K before using new T-calibration) and 124 K (115 K before using new T-calibration)”, we were referring to the parameter D . We did not intend to claim that the diffusive part is dominant, but rather that the associated decay rate increases between 121 K and 124 K. We updated the text for improved clarity on page 9:

With increasing temperature, we observe that diffusion coefficient (D) rises sharply (i.e., the decay rate increases) between 121 K and 124 K, while the ballistic velocity (v) remains almost constant (Fig. 6, lower right panel).

From the fits, the diffusive part concerns only the long-time decay of the data, which is highly sensitive to the baseline determination and which at higher temperatures evolves to a crystalline contribution. At 115K, crystalline contributions are already visible in the WAXS and could lead to an extra static contribution. How can the authors exclude an influence of the crystallization at 115K? A better discussion would definitely help.

Crystalline contributions can result in a static contribution to g_2 . We have observed this recently in a study at ambient pressure, where the static contribution resulted in a heterodyne signal (Li et al. 2023, ref. 14). However, we do not observe such a heterodyne signal here. At higher temperatures (see sample B at $T_{\text{sample}} = 145$ K) a double exponential decay is visible, which can be assigned to the contribution from crystalline ice. This is discussed now in SI, see Figure S10. In addition, the determination of the baseline is further discussed in detail, as written above and shown in Figure S11.

-figure 7 is not convincing. G and F are added although pressure is indirectly estimated and no information on flux (and thus on temperature) is reported. If there's an uncertainty on both parameters, they cannot be used.

We removed the data points G and F .

The authors say << The diffusive dynamics of sample B at 130 K defines the glass transition temperature T_g (black star). Are the authors referring to $T_{\text{cryo}}=130$ K or $T_{\text{cryo}}=115$ K? It would be easy to report directly the corrected temperatures and just cite the methods for the

calibration. If they refer to $T_{\text{cryo}}=130\text{K}$ the dynamics is not reported in the article, if they refer to $T_{\text{cryo}}=115\text{K}$, which should be the case, I don't see how they can claim to have diffusive dynamics at the light of my previous comment. A discussion on why the dynamics at T_g is expected to be diffusive is also desired.

We are sorry for the confusion and apologise to further confusion due to the new temperature calibration (see previous discussion in the reply letter).

We referred to $T_{\text{sample}} = 130\text{K}$, hence $T_{\text{cryo}} = 115\text{K}$. To improve readability, we changed all temperatures to the corrected ones throughout the manuscript, taking a new calibration into account. This is $T_{\text{sample}}(\text{new}) = 124\text{K}$.

As described above, we observe that diffusion coefficient (D) rises sharply (i.e., the decay rate increases) between $T_{\text{sample}}(\text{new}) = 121\text{K}$ and 124K , while the ballistic velocity (v) remains almost constant (Figure 6, lower right panel). This analysis is where we base our interpretation on. We also state that the diffusion coefficient is still small and no pure Brownian motion was observed. This behaviour is also expected in the vicinity of the glass transition temperature.

- pressure is measured only at the beginning although it's known to increase with temperature. How the authors know that is constant? The pressure stability and temperature stability during the measurements are not much discussed

We measured the pressure change upon cooling in a separate experiment from room temperature to 80K . The pressure in the temperature range $80 - 130\text{K}$, which is where the experiments are conducted, was observed to change no more than 10% .

We added the following sentence on page 13:

Also note that the pressure is measured at the lowest temperature. During heating inside the DAC the P - T pathway will not be isobaric. We expect a small decrease in pressure ($\sim 10\%$) within the investigated temperature range. Since the given pressure anyway carries a large error bar, we relate to the initial pressure throughout the manuscript. In-situ pressure determination during the XPCS experiments was not possible for technical reasons (see SI).

We would like to thank all reviewers for their encouragement, detailed comments and additional feedback provided. We appreciate the opportunity to clarify some points and improve our work further.

Reviewer #1:

The authors have addressed in detail the points raised in my previous report. I appreciate the clarification on the kinds of transformations observed upon heating (eg, HDA LDA, HDA HDL LDA/LDL) in page 10 (Conclusions), and the new Fig. 7. I find the new version of the manuscript suitable for publication in Communications Chemistry.

We would like to sincerely thank Reviewer 1 for the positive assessment.

Reviewer #2:

Although the revision has shown some improvements, I find some of the responses to my previous comments not satisfactory, as listed below.

Pressure measurement: The authors seem to insist the validity of a pressure of 0.08 GPa by ruby fluorescence. Note that this 0.08 GPa corresponds to a ruby line shift of only ~ 0.003 nm, which is comparable to (even a little smaller than) the uncertainties as shown in the Fig. 1C, where the errors (from fitting statistics alone) in their fluorescence measurements are listed as ± 0.003 nm and ± 0.004 nm. This means that the employed ruby technique (and the spectrometer Ocean Optics HR4000) does not provide necessary precision to reliably measure a pressure as small as 0.08 GPa. In addition, I am troubled by the added error bar of ± 0.02 GPa in the revision, because the authors do not provide any basis of this estimate.

We thank the editor and referee to let us clarify these points. In the following we explain how we estimated the pressure and related error bars. We agree that ruby fluorescence is not optimal in this pressure range. We hope that the new Figure S1 (see below) and the extended error bar discussion make the limitations clear for the reader. We further added on page 13:

The shift of the ruby line is very small at the low pressures investigated here, a detailed discussion can be found in SI. We admit that this leads to a quite large error bar. Future experiments might consider to use alternative pressure gauge {ref. Shen et. 2021}.

We still believe that the value of 0.08 GPa is the best estimate for the pressure, taking both WAXS and ruby measurements into account. Let us first summarize some arguments before coming back to the ruby measurements.

- All eHDA samples were prepared externally, following the same protocol, and later cold loaded to the DAC. The samples were quench- recovered from 0.08 GPa.
- Literature (e.g. Nelmes et al.) shows that the first diffraction maximum of eHDA would shift towards larger Q if the sample is pressurized beyond 0.1 GPa. This is seen for sample D, as a Q position of 2.2 \AA^{-1} can only be explained by higher pressure (Fig. S2).
- As seen and discussed in Fig. S2 and our previous reply, samples A and B cannot have a pressure higher than 0.08 GPa.
- Also the temperature range where we observe LDA in this study, compared to our ambient pressure studies, is much smaller. According to Seidl et al. this happens in a pressure range slightly below 0.1 GPa.
- We are left with the question of how much pressure it is between ambient and 0.08 GPa.
- We apologize that the provided error bar was not well motivated

According to the KANTOR website (ref. 36 main manuscript), which includes the calibration by Shen et al. 2020 and temperature correction by Datchi et al. 2007, the R1 ruby line is shifting by 0.365 nm / GPa (Datchi et al.). A similar value is reported in the new publication (Shen et al. 2021), where the linear coefficient ($\Delta\lambda/\Delta P$) of the pressure dependence of the R1 line is given as 0.3722 nm/GPa. This corresponds to a shift of approximately 0.03 nm for 0.08 GPa. Even if this is in the order of the resolution of our Ocean Optics spectrometer, the FWHM of 0.15 nm of the R1 line allows us to distinguish between location of the R1 line at 0 GPa and 0.08 GPa. At the lowest sample temperature of 93 K, the Ruby line would be centred at 693.335 at 0 GPa and 693.366 at 0.08 GPa. This is a difference of 0.033 nm compared to our 0.003-0.004 nm error bars from curve fitting. We can clearly discriminate between these two values. Hence, the pressure must be higher than 0 GPa.

We measured several ruby fluorescence spectra during the beamtime. Below (left panel) are three ruby lines of sample B is presented. This is now shown in Fig. S1-A, including error range estimation (S1-B). We also replaced the Ruby line in Fig. 1 by the more representative middle curve. Fig. S1 in addition contains the Ruby measurement for sample D for comparison. The error bar of 0.02 GPa covers the range of 0.07-0.1 GPa in sample B.

Note: At 693.366 nm, a temperature uncertainty of ± 4 K would affect the pressure by 0.01 GPa. Taking also the error bars of the three measurements into account, this all lies within the range of ± 0.02 GPa.

Fig. for reviewer.

similar to new Fig. S1-A,B

We added the following on page 1-2 of SI:

For sample B, main sample of the study, we measured the ruby spectrum three times, as shown in Fig. S1A. Spectra were fitted with a Gaussian function (scipy curve fit) to determine the peak maximum with an error of 0.003-0.004 nm. This corresponds to a pressure range of 0.07-0.1 GPa for sample B (Fig. S1B), centered around a mean peak position of 693.366 nm, equivalent to 0.08 GPa at 93 K. Ruby reference of the Ruby sphere was measured at ambient pressure at 296 K to be 694.21 nm.

Furthermore, when discussing the pressure drift as the cell temperature was changed, a 10% decrease in pressure was given/added in the revision, again without any support from measured numbers.

We ran several heating tests with a pre-pressurised DAC before and after the beamtime. We observed in all of them a 3.5%- 8 % pressure decrease per 10 K. Below we present two runs for the lower pressure range around 0.2 GPa and higher pressures at around 1-2 GPa. For those measurements glycerol has been loaded to the DAC. A third measurement is shown in SI, where low-density amorphous ice has been cold-loaded to the DAC, similar to the reported XPCS experiments. The pressure development during the transition from eHDA to LDA(L) was not measured, but we assume here, that the pressure change prior to the transition would behave similar to what we observed for LDA and glycerol. All values are located within the estimated error bar of 0.02 GPa.

Fig. for reviewer

The following text was added to SI on page 2:

We did independent measurements after the beamtime, both on the pressure evolution as well as on the sample temperature (see below). Fig. S1C shows three datapoints for low-density amorphous ice cold-loaded to the DAC and the related pressure change upon heating. The pressure decreases by ~3.5% by 10 K. In addition, we did the same test loading glycerol to the DAC in two different pressure ranges, and observe a percentual pressure change of 6-8% pressure decrease per 10 K step. In-situ pressure determination during the XPCS experiments was not possible for technical reasons. Using an average value, the pressure during our experiment might decrease at 123 K down to 0.071 GPa, representing a 11% pressure reduction during our measurements. This lies within the range of 0.08 ± 0.02 GPa.

New Fig. 1-C: C) shows an external measurement (after the beamtime) of cold-loaded LDA, to determine the pressure change upon heating.

Main manuscript, page 13, changed to : (*~11% in the range 93K - 123 K, see Fig. S1-C*).

The authors argue that the ruby fluorescence "was the best choice". But that choice does not justify the reported pressure values. In fact, the use of fluorescence of f-electron metals, e.g., Sm²⁺:SrFCl (<https://doi.org/10.1080/08957959.2021.1931168>) provides >3 times better resolution at low pressure.

We thank the referee for pointing on this publication. Unfortunately, we overlooked this before. Also, the publication appeared after our first experiments. We will explore the new possibilities in our future experiments. However, the above mentioned study does not provide calibration data for liquid nitrogen temperature.

We added the following on page 13:

The shift of the ruby line is very small at the low pressures investigated here, a detailed discussion can be found in SI. We admit that this leads to a quite large error bar. Future experiments might consider to use alternative pressure gauge {ref. Shen et al.. 2021}.

On the other hand, x-ray data (FSDP, WAXS, transition) may give better precision in pressure determination in this study. The authors state that "Note, we do not use the crystalline Bragg peaks for calibration. The equation of states for hexagonal ice indicates that a peak shift in the relevant pressure range is very small ($< 0.005 \text{ \AA}^{-1}$) and therefore not detectable with our WAXS detector arrangement". This argument is invalid. To some degree, this argument applies better to the ruby technique in this case. In proper WAXS setups, such as the one reported in this study, a detectable limit should be better than 0.001 \AA^{-1} . The mentioned 0.005 \AA^{-1} range of peak shift not only should be detectable, but can be used for pressure determination. Generally speaking, no fluorescence technique can provide better precision in pressure determination than the x-ray diffraction technique.

We agree with the referee that, in general, X-ray diffraction can be a precise method for pressure determination. However, we want to point out, that the here used WAXS setup is not installed in order to gain high precision, but to gain information on the amorphous ice transition and crystallization events. Beside the fact that at beamline P10, no DAC experiment has been performed prior to this experiment, the experiment was mainly designed for XPCS in SAXS geometry. The WAXS detector does not belong to the standard configuration of the beamline. The second detector was added only for this experiment and mounted sideways on an additionally clamped breadboard at a fixed position (15 cm distance). This WAXS detector is tilted with respect to the beam path (see sketch below). It is further not motorized and only covers a small part of the diffraction ring. Calibration was done using LaB₆ at ambient pressure as well as inside a closed, but not pressurised, DAC. However, the cold-loaded DAC is inserted to the cold finger inside a large liquid nitrogen bath, see figure S3-B₁, and the position of the DAC might vary by up to 1 mm. In the following we show that at large diffraction angles

and close sample-detector distances, the geometry calibration is highly sensitive to displacements of the sample along the beam direction. The sketch below shows how a ray direction may be mischaracterized if the sample position is known to some precision $\pm \Delta x$. This error is the largest at the point of normal incidence between diffracted rays and the detector plane. We estimate that the sample position is reproducible to within $\pm 1 \text{ mm}$, resulting in a maximum uncertainty of $\pm 0.013 \text{ \AA}^{-1}$ in momentum transfer (the error

on diffraction angles is found using the law of cosines). This calibration error is small enough for discrimination between phases of amorphous ice, but not for determination of pressure from crystalline ice peaks.

The following is added to page 12 (cryo-loading procedure):

The DAC was mounted on a customized copper holder by sliding into the Cu-block (Fig. S3-B₁) inside a LN₂ bath. This might lead to a positional uncertainty of ±1mm.

and to page 13:

LaB₆ was used for calibration. However, the WAXS detector is tilted with respect to the beam path and we estimate the uncertainty in Q to be ± 0.013 Å⁻¹ for the WAXS measurements.

Temperature offset: The response of the temperature offset is also not satisfactory. My comment was "Because the offset is clearly dependent on thermal pathways (cooling, heating, and their rates), it is not a fixed number, but should vary in a range from positive to negative depending on the pathways and the experimental setup". The authors only responded the effect of cooling rate (while their experiments were conducted under heating cycles). The authors mentioned (correctly) that "the thermocouple (sample temperature) lags behind", which means that the sample temperature could be higher than the cryostat temperature during cooling, and that the opposite could occur during heating cycle. That is the sample temperature could be lower than the cryostat temperature. If this is indeed the case, then the temperature correction should be opposite. The interpretation in the main text regarding the glass transition temperature should then be completely revised.

We fully agree with the referee that the offset depends on the thermal pathway. This is why we equilibrate the sample for 20-30 min before each run. We apologise for the confusion and for not being clear enough. Table S2 in the previously submitted SI, showed the measured heating data, also for the external thermocouple measurement (*previous right columns, runs 1 and 2*). The two runs had been done after the beamtime, showing selected and equilibrated datapoints.

We here attach two new temperature measurements (runs 3 and 4), where both thermocouple (yellow) and diode A (blue) have been recorded using the same PID controller (Lakeshore 335). These two measurements were done using the same cryostat, and show heating in run 3 due to a low flow-rate in the nitrogen supply (time span 1-2 h), and a noise in the feedback loop in run 4 (time span 1.5 -3h). The data clearly show how the thermocouple (=T_{sample}) follows the diode temperature. The graphs visualizes the offset after 30 min equilibration time being rather constant, and that the sample temperature after equilibration is never lower than the measured values by diode A. An exception appears for a heating step > 10 K (run 4, at 7h), which we never exceeded in the XPCS experiment. During the beamtime we **always** waited until both diodes A and B stabilized for 20 – 30 min.

We show this plot for reviewing purpose only. We add a magnification of plot A (run 3) to SI Fig. S4-A. The average-offset after 30 min is plotted in Fig. S4-B. The obtained linear relation between the two sensors has been used to estimate the sample temperature, as listed in table S2. The values differ by only 1-2 K compared to the previous version of the manuscript, as the offset become slightly smaller at higher T.

Fig. for reviewer

The following was added to SI on page 4:

Fig. S4-A shows an external measurement for diode A and a thermocouple glued to the gasket (see above). The plot shows that the sample temperature (thermocouple) lags behind but reaches a plateau after 20-30 min, which is when we started the XPCS experiment. In total we did four external heating runs, each with a freshly glued thermocouple, the averaged data are summarized in Figure S4-B. We estimate the temperature offset during the XPCS experiment using the equation shown in Fig. S4-B, the values are provided in table S2.

Fig. S4: External measurement with thermocouple (tc) glued to gasket, heating in steps of 10 K and equilibrated for 30 min. B shows the offset between thermocouple and Si-diode A after 30 min. The determined function can be used to estimate the sample temperature during the XPCS experiment.

Reviewer #3:

I thank the authors for the effort they made to reply to all comments and improve the quality of their work. However, at the light of the responses to all referees and the new version of the manuscript, I am still not convinced on the data interpretation and accuracy. This is an important article that can pave the way to numerous experimental and numerical studies. I recognize the investment of the authors and their expertise in the field of amorphous water. Still, the argumentation employed by the authors to determine the glass transition temperature is not convincing in my opinion. I cannot therefore support the publication of the article. The reproducibility of the data of sample B in a second experiment, together with a better control on the pressure, would definitely help elucidating the origin of the observed dynamics.

We appreciate the reviewers acknowledgement of the potential significance of our study. Although conducting an additional experiment on Sample B would provide further insights, we are currently limited by the nature of synchrotron experiments, which require specialized facilities and are limited by restricted access time, conducting a second experiment within the current project scope is unfortunately not feasible. To ensure the reliability of our current data, we have employed rigorous experimental protocols. We have now added a detailed discussion about the determination of our uncertainties in pressure and temperature (SI new Fig. S1 and Fig. S4). We also refer to our reply below, regarding sample A, with respect to reproducibility of the XPCS data. In the manuscript we've briefly outlined plans for future studies to follow up on these results under more controlled conditions. This includes a pressure gauge which is more suitable for the lower pressure range as well as a different type of diamonds. We hope this helps clarifying the current limitations of our study while still showing the value of our findings as an initial investigation.

Scattering Pattern: I still believe that the speckle pattern in Fig. S2C has an additional contribution beyond the Kossel line, which arises from a second beam reflected by the diamonds through the sample. The Kossel line cannot explain the additional speckles in the 3rd and 4th modules with a circular pattern not centred with respect to the direct beam (and a shape similar to the scattering of the sample). As the article does not mention whether this scan corresponds to sample A or B, it is impossible to know the effect on the dynamics. I suggest the authors to verify that all scans used to calculate the dynamics are clean, with only Kossel lines (as in the new Fig. 1 where the authors did the azimuthal test).

We are very sorry for the confusion, the detector image shown in Figure S2-C (now S3-C) belongs to sample D, measured at a pressure of ~0.75 GPa. We previously mentioned this in the text under section "detector images", but forgot to add this information in the figure caption. This is done now. Due to the strong pseudo-Kossel line, the sample was not analysed. We tried to mask the area, and alternatively analyzed only modules 5-8, however, the obtained g2 curves were either too noisy or showed no decay. We exclusively show the detector image of sample D, in order to point out why the XPCS data at 0.75 GPa had not been analysed, and to add the WAXS data of this high-pressure sample D. We apologise that this did not become clear in the text and we have now highlighted this in the following sections:

We added the following text to figure caption S3:

...of sample D with strong Pseudo-Kossel line, or another scattering artefact.

We added to caption of Table S1:

The analysis of the correlation function at the highest pressures (sample D) was not possible, due to the appearance of Pseudo-Kossel lines (see section “Experimental setup”).

You can find this in the section “detector images” on SI on page 6:

Sample D could therefore not be analyzed in terms of dynamics. For lower pressures as in samples A and B, weak Pseudo-Kossel lines can be masked out, without influencing the result. No effect of directionality has been found. Figure S6 shows the masked region on the detector, and g_2 curves calculated independent from the upper and lower part of the detector.

Glass transition identification: I still don't understand how the induced dynamics of sample A which is influenced by a structural damage can be used to strengthen the validity of the measured dynamics of sample B.

We want to point out that the X-ray induced effect does not cause a structural damage, it mainly results in an additional heating, hence the actual sample temperature is higher than expected from the thermocouple estimation. A structural damage would be seen by immediate phase transition or crystallization, which here does not happen at the lower temperatures. Regarding the previous question, this is why showing sample A supports also the finding in sample B, approximately one can say, “simply the temperature scale is off”. That's maybe too much oversimplified, however, we refer to studies on protein solutions (Bin et al. JPBC 127, 2023) showing that normalizing the correlation function g_2 time axis with respect to the flux density is possible. Also studies on oxide glasses showed that while the structure does not depend on the flux, strong fluxes can induce a non-trivial microscopic but reversible motion (Ruta et al. Scientific reports 7 (2017)).

We added the following text on page 10:

Such acceleration effects, without structural damage, have also been observed in hydrated proteins⁴⁰, where normalizing the g_2 time axis with respect to the flux density was possible, but cannot be applied here, due to the narrow temperature window.

Furthermore, even if the motion is more diffusive in sample B at high temperature, the strength of this process is reduced, indicating that this process is less dominant, so I don't see how this can be attributed to the glass transition, and why this aspect is not commented in the text.

The nature of the ballistic component can be manifold. One possible explanation relates to earlier findings at ambient pressure, where the ballistic component seems to be related to the growth of static LDA domains, strain release or growth of crystalline domains might be another origin. Our fits show that the rate of the ballistic velocity (v) remains almost constant. Both the single exponential KWW decay fit as well as the multi-fit approach lead to similar diffusion coefficients.

We add the following to the text on page 10:

While the velocity of the ballistic component remains almost constant (Fig. 6), the strength of the ballistic component grows. The nature of the ballistic component can be manifold. One possible explanation relates to earlier findings at ambient pressure, where the ballistic component seems to be related to the growth of static LDA domains, as the transition to LDA also involves a volume change of almost 20%.¹⁴ Strain release or growth of crystalline domains might be another origin. While the domains grow in size, this might explain the increasing strength of the ballistic component with constant velocity.

As written in my previous report, I would appreciate also if the authors can provide an evidence of the fact that the dynamics at T_g should become diffusive. There are different theoretical works in glass formers suggesting a completely different mechanism of particle motion in supercooled liquids (see for instance Bhattacharyya, J. Chem. Phys. 132 104503, 2010). This is still not commented.

The nature of glass forming liquids is still highly debated topic in literature, not only in water research, but also else. This is a follow up study on our work on amorphous ice at ambient pressure, where we used the same definitions in order to determine the glass transition temperature (Perakis et al. PNAS 114, 2017). The relation of diffusive motion related to the glass transition is also describe in refs. 12, 24 or e.g. Conrad et al. (*Dynamics in glass forming polymers, PRE 91, 2015*). To make this more clear for the reader we added the following to the main text:

page 2:

The nature of the glass transition in supercooled liquids, not only water, is highly debated and different mechanism are discussed in literature.^{38,39} A combination of different tools and approaches is needed to shed light to this debate.

page 4:

In our analysis we relate the glass transition temperature to the above describe KWW exponent γ as well as to the Stokes-Einstein diffusion coefficient^{24,12} D , following our previous approach at ambient pressure.¹³⁻¹⁵

Furthermore, at 136K the “The WAXS image at this temperature shows a small amount of crystalline ice, but no drastic growth”. It is well known that dynamic probes are more sensitivity that single point correlators. How can the authors exclude that crystallization already affect the dynamics at 130K? This would indeed explain why the strength of the diffusion process decreases.

As mentioned in the text and described above (see previous answer), growth of crystalline domains might indeed be one reason for the ballistic domains.

We add the following to the text on page 10:

*While the **velocity of the ballistic component** remains almost constant (Fig. 6), **the strength of the ballistic component grows**. The nature of the ballistic component can be manifold. One possible explanation relates to earlier findings at ambient pressure, were the ballistic component seems to be related to the growth of static LDA domains, as the transition to LDA also involves a volume change of almost 20%.¹⁴ Strain release or growth of crystalline domains might be another origin. **While the domains grow in size, this might explain the increasing strength of the ballistic component with constant velocity.***

"With which criterion the temporal binning for the correlation functions are selected?" With this question i was meaning the number of frames used to get correlation curves. This is completely arbitrary and not enough commented in the text. As the TTC are not stationary, this choice can dramatically affect the results.

The XPCS measurements in this work were measured for 1000 s with 1 s exposure times per frame, therefore our data set contains 1000 frames. Due to the instability of sample A, only the first 300 frames of the measurements were used for the correlation analysis. Looking at the TTC of sample B, we observed heterogeneous dynamics, as reviewer 3 pointed out. Therefore, only the first 500 s (frames) were used to obtain correlation curves, since the dynamics seem to remain rather stable during this period.

I am still confused by the pressure determination and stability. There is an uncertainty of 10% as pressure is measured only at the beginning without considering temperature effects. Then there is an uncertainty due to the fact that pressure is checked by the ruby luminiscence spectra which is challenging for low pressures values. Finally there is an uncertainty due to the fact that pressure is calibrated from the evolution of the amorphous structure measured in a set-up which is not optimized for diffraction (and without using a crystalline calibrant). At the light of all this, how is possible to have "The error in pressure determination is at least ± 0.02 GPa"? This value seems to low for this set-up. How this is estimated? More importantly, pressure should remain constant during the measurements. How can the authors be sure about it?

We refer to our reply to referee #2 above. We now added a detailed description of how the error bar of 0.02 GPa was estimated to the SI. This includes as new Fig. S1, where S1-A shows three independent ruby measurements of sample B, the related pressure range in S1-B and the effect of heating (S1-C). At the average value of 693.366 nm at 93 K, a temperature uncertainty of ± 4 K would affect the pressure by 0.01 GPa. Taking also the error bars of the three measurements into account, this all lies within the range of ± 0.02 GPa. A better documented temperature estimation is now provided in Fig. S4.

New: Fig. S1: (A) shows R1 lines by three independent Ruby measurements during the beamtime of sample B as well as on sample D (red). (B) shows the related peak position of sample B and the derived pressure, showing the uncertainty of the pressure determination. (C) shows an external measurement (after the beamtime), to determine the pressure change upon heating.

New: Fig. S4: External measurement with thermocouple (tc) glued to gasket, heating in steps of 10 K and equilibrated for 30 min. B shows the offset between thermocouple and Si-diode A after 30 min. The determined function can be used to estimate the sample temperature during the XPCS experiment.

Dear Editor,

we would like again to thank all reviewers for their encouragement and valuable feedback, and are happy that our manuscript got accepted.

We here attach the final revision following the request by reviewer #2

Reviewer #2:

In my previous review, I mistakenly put a shift of 0.003 nm (instead of 0.03 nm) for a pressure of 0.08 GPa. I thank the authors for clarifying this and the added information in the revision. I also find the responses to the other comments reasonable. In my opinion, the revised manuscript is ready for publication with minor modification (see below).

On the temperature offset, it appears that there is always a large temperature difference between the thermocouple readings and diode readings, even after a few hours (see figs. S3-D and S4). Initially I assumed that these two temperatures (from thermocouple and diode) suppose to be close within a few degrees after equilibrium is reached. Apparently their measured results before and in more recent repeated experiments show that thermocouple readings are consistently at higher temperatures. This could be related to the cryostat design, the sensor calibration, the heater location, etc. A note on this would help readers to understand the positive offset corrections in the heating cycles.

In order to enable the readers to better find the information about the described temperature offset, we added a new paragraph in the method section named "Cryostat and sample temperature" (marked yellow on page 12, blue words have been added and the previous text reshuffled).

Cryostat and sample temperature. We used a JANIS ST-400 cryostat (run with LN₂). Two Si-Diodes are placed at the bottom of the cold finger, one buried inside the cold finger close to the heater element (Diode A), as provided by the company, and a second Diode B mounted on the copper holder (Fig. S3). Both diodes were used to monitor the temperature throughout the experiments. Due to the current cryostat / sample-holder design, a small gap in the order of several 100 micrometers remains between the Cu-sample holder and the DAC, which cannot be avoided when cold loading the DAC (Fig. S3-B₁). The temperature of the sample is therefore consistently higher than the reading of the two diodes. We estimate an offset of about 12-15 K between the measured temperature (Diode A) and that of the sample inside the DAC by using an additional thermocouple. Temperatures given throughout the manuscript are offset-corrected. Details to this temperature calibration are described in the SI, see Fig. S4, Fig. S5 and Table S2.

In *XPCS at elevated pressure and cryogenic temperatures: Multicomponent dynamics in amorphous ice* Karina and co-authors investigate the dynamics of amorphous ice at 0.08GPa during its transformation from a HDA to LDA. Water has a complex and debated phase diagram with many crystalline phases and at least two amorphous ice solids phases. These amorphous phases are fundamental in many cosmological processes and have been often connected to the well-known anomalies of liquid water. In this technical article, the authors present a challenging experiment to measure the glass transition of amorphous ice at elevated pressures and cryogenic temperatures from time-resolved speckle autocorrelation measurements of x-rays scattered by the sample. The authors claim to see the presence of two dynamical contributions, a ballistic and a diffusive one, during the HDA-LDA transition and they located the glass transition temperature of LDA. This is an extraordinary claim, however the evidence provided for it raises several crucial questions that I think the authors should address

- The SAXS profile presents an additional contribution which is discussed in terms of pseudo Kossel lines. It looks to me that alternative explanations could also be considered and discussed. A similar pattern could for instance arise from a multiple scattering of the sample scattering, superimposed on the first one, probably reflected by a diamond acting as an analyser. This could explain the partial scattered profile, centred at a different position which is cut by the misalignment of the diamond. A simple rotation of the cell should verify this hypothesis. The presence of a secondary SAXS pattern could explain the “interaction with the speckles patterns” even outside the “masked pixels”. Have the authors compared the analysis of the bottom part of the SAXS profile with the top part? Is it the same? This aspect is fundamental to assess the reliability of the results.

- For what discussed in the work, is clear that Sample A suffers of radiation damage: the transition is at the wrong temperature, wrong dynamics, already at 82K in figure S4 the WAXS structure is not stable and changes with time, and even the SAXS $I(Q)$ has a different profile in the two samples. After the transition, the $I(Q)$ of sample A is basically a straight line in log scale. So, as the authors as well are considering sample A not reliable, I don't understand why the measured dynamics is still discussed and considered partially representative of the transition HDA- LDA. It could be directly moved to SI. the data of sample A are all measured on a same spot and at 104K on a second position. As the beam is locally heating or damaging, the two datasets cannot be compared having different doses.

- Data analysis lacks of some information. How are the baselines determined? Is the contrast constant during the transition? With which criterion the temporal binning for the correlation functions are selected? Looking at figure 6, not all data sets are faster after 500s. Although the authors claimed the opposite, the strength of the diffusive process decreases at higher temperature (A is maximum at 115K). How can this process be dominant if its strength is so low? From the fits, the diffusive part concerns only the long-time decay of the data, which is highly sensitive to the baseline determination and which at higher temperatures evolves to a crystalline contribution. At 115K, crystalline contributions are already visible in the WAXS and could lead to an extra static contribution. How can the authors exclude an influence of the crystallization at 115K? A better discussion would definitely help.

- figure 7 is not convincing. G and F are added although pressure is indirectly estimated and no information on flux (and thus on temperature) is reported. If there's an incertitude on both parameters, they cannot be used. The authors say << The diffusive dynamics of sample B at 130 K defines the glass transition temperature T_g (black star)>>. Are the authors referring to $T_{cryo}=130K$ or $T_{cryo}=115 K$? It would be easy to report directly the corrected temperatures and just cite the methods for the calibration. If they refer to $T_{cryo}=130K$ the dynamics is not reported in the article, if they refer to $T_{cryo}=115 K$, which should be the case, I don't see how they can claim to have diffusive dynamics

at the light of my previous comment. A discussion on why the dynamics at T_g is expected to be diffusive is also desired.

- pressure is measured only at the beginning although it's known to increase with temperature. How the authors know that is constant? The pressure stability and temperature stability during the measurements are not much discussed